# Identification of putative causal loci in whole-genome sequencing data via knockoff statistics

Zihuai He [1,2✉], Linxi Liu [3], Chen Wang [4], Yann Le Guen [1], Justin Lee[2], Stephanie Gogarten [5], Fred Lu[6], Stephen Montgomery [7,8], Hua Tang [6,7], Edwin K. Silverman[9], Michael H. Cho [9], Michael Greicius[1] & Iuliana Ionita-Laza[4✉]

The analysis of whole-genome sequencing studies is challenging due to the large number of rare variants in noncoding regions and the lack of natural units for testing. We propose a statistical method to detect and localize rare and common risk variants in whole-genome sequencing studies based on a recently developed knockoff framework. It can (1) prioritize causal variants over associations due to linkage disequilibrium thereby improving interpretability; (2) help distinguish the signal due to rare variants from shadow effects of significant common variants nearby; (3) integrate multiple knockoffs for improved power, stability, and reproducibility; and (4) flexibly incorporate state-of-the-art and future association tests to achieve the benefits proposed here. In applications to whole-genome sequencing data from the Alzheimer's Disease Sequencing Project (ADSP) and COPDGene samples from NHLBI Trans-Omics for Precision Medicine (TOPMed) Program we show that our method compared with conventional association tests can lead to substantially more discoveries.

[1] Department of Neurology and Neurological Sciences, Stanford University, Stanford, CA, USA. [2] Quantitative Sciences Unit, Department of Medicine, Stanford University, Stanford, CA, USA. [3] Department of Statistics, Columbia University, New York, NY, USA. [4] Department of Biostatistics, Columbia University, New York, NY, USA. [5] Department of Biostatistics, University of Washington, Seattle, WA, USA. [6] Department of Statistics, Stanford University, Stanford, CA, USA. [7] Department of Genetics, Stanford University, Stanford, CA, USA. [8] Department of Pathology, Stanford University, Stanford, CA, USA. [9] Channing Division of Network Medicine and Division of Pulmonary and Critical Care Medicine Division, Brigham and Women's Hospital, Harvard Medical School, Boston, MA, USA. ✉email: zihuai@stanford.edu; ii2135@cumc.columbia.edu

The rapid development of whole-genome sequencing technology allows for a comprehensive characterization of the genetic variation in the human genome in both coding and noncoding regions. The noncoding genome covers ~98% of the human genome, and includes regulatory elements that control when, where, and to what degree genes will be expressed. Understanding the role of noncoding variation could provide important insights into the molecular mechanisms underlying different traits.

Despite the increasing availability of whole-genome sequencing datasets including those from moderate to large scale projects such as the Alzheimer's Disease Sequencing Project (ADSP), the Trans-Omics for Precision Medicine (TOPMed) program etc., our ability to analyze and extract useful information from these datasets remains limited at this point and many studies still focus on the coding regions and regions proximal to genes[1,2]. The main challenges for analyzing the noncoding regions include the large number of rare variants, the limited knowledge of their functional effects, and the lack of natural units for testing (such as genes in the coding regions). To date, most studies have relied on association testing methods such as single variant tests for common variants, gene-based tests for rare variants in coding regions, or a heuristic sliding window strategy to apply gene-based tests to rare variants in the noncoding genome[3,4]. Only a few methods have been developed to systematically analyze both common and rare variants across the genome, owing to difficulties such as an increased burden of the multiple testing problem, more complex correlations, and increased computational cost. Moreover, a common feature of the existing association tests is that they often identify proxy variants that are correlated with the causal ones, rather than the causal variants that directly affect the traits of interest. Identification of putative causal variants usually requires a separate fine-mapping step. Fine-mapping methods such as CAVIAR[5] and SUSIE[6] were developed for single, common variant analysis in GWAS studies, and are not directly applicable to window-based analysis of rare variants in sequencing studies.

Methods that control the family-wise error rate (FWER) have been commonly used to correct for multiple testing in genetic associations studies, e.g., a $p$-value threshold of $5 \times 10^{-8}$ based on a Bonferroni correction is commonly used for genome-wide significance in GWAS corresponding to a FWER at 0.05. The number of genetic variants being considered in the analysis of whole-genome sequencing data increases substantially to more than 400 million in TOPMed[2], and FWER-controlling methods become highly conservative[7]. As more individuals are being sequenced, the number of variants increases accordingly. The false discovery rate (FDR), which quantifies the expected proportion of discoveries which are falsely rejected, is an alternative metric to the FWER in multiple testing control, and can have greater power to detect true positives while controlling FDR at a specified level. This metric has been popular in the discovery of eQTLs and Bayesian association tests for rare variation in autism spectrum disorder studies[8–11]. Given the limited power of conventional association tests for whole-genome sequencing data and the potential for many true discoveries to be made in studies for highly polygenic traits, controlling FDR can be a more appealing strategy. However, the conventional FDR-controlling methods, such as the Benjamini-Hochberg (BH) procedure[12], often do not appropriately account for correlations among tests and therefore cannot guarantee FDR control at the target level, which can limit the widespread application of FDR control to whole-genome sequencing data.

The knockoff framework is a recent breakthrough in statistics to control the FDR under arbitrary correlation structure and to improve power over methods controlling the FWER[13,14]. The main idea behind it is to first construct synthetic features, i.e., knockoff features, that resemble the true features in terms of the correlation structure but are conditionally independent of the outcome given the true features. The knockoff features serve as negative controls and help us select the truly important features, while controlling the FDR. Compared to the well-known Benjamini-Hochberg procedure[12], which controls the FDR under independence or a type of positive-dependence, the knockoff framework appropriately accounts for arbitrary correlations between the original variables while guaranteeing control of the FDR. Moreover, it is not limited to using calibrated $p$-values, and can be flexibly applied to feature importance scores computed based on a variety of modern machine learning methods, with rigorous finite-sample statistical guarantees. Several knockoff constructions have been proposed in the literature including the second-order knockoff generator proposed by Candès et al.[14] and the knockoff generator for Hidden Markov Models (HMMs) proposed by Sesia et al.[15,16]. The HMM construction has been applied to phased GWAS data in the UK biobank. However, these constructions can fail for rare variants in whole-genome sequencing data whose distribution is highly skewed and zero-inflated, leading to inflated FDR. Romano et al.[17] proposed deep generative models for arbitrary and unspecified data distributions, but such an approach is computationally intensive, and therefore not scalable to whole-genome sequencing data.

Our contributions in this paper include a sequential knockoff generator, a powerful genome-wide screening method, and a robust inference procedure integrating multiple knockoffs. The sequential knockoff generator is more than 50 times faster than state-of-the-art knockoff generation methods, and additionally allows for the efficient generation of multiple knockoffs. The genome-wide screening method builds upon our recently proposed scan statistic framework, WGScan[18], to localize association signals at genome-wide scale. We adopt the same screening strategy, but incorporate several recent advances for rare-variant analysis in sequencing studies, including the aggregated Cauchy association test to combine single variant tests, burden and dispersion (SKAT) tests, the saddlepoint approximation for unbalanced case-control data, the functional score test that allows incorporation of functional annotations, and a modified variant threshold test that accumulates extremely rare variants such as singletons and doubletons[19–26]. We compute statistics measuring the importance of the original and knockoff features using an ensemble of these tests. Feature statistics that contrast the original and knockoff statistics are computed for each feature, and can be used by the knockoff filter to select the important features, i.e., those significant at a fixed FDR threshold. The integration of multiple knockoffs further helps improve the power, stability, and reproducibility of the results compared with state-of-the-art alternatives. Using simulations and applications to two whole-genome sequencing studies, we show that the proposed method is powerful in detecting signals across the genome with guaranteed FDR control.

Our knockoff method can be considered a synthetic alternative to knockout functional experiments designed to identify functional variation implicated in a trait of interest. For each individual in the original cohort, the proposed method generates a synthetic sequence where each genetic variant is being randomized, making it silent and not directly affecting the trait of interest while preserving the sequence correlation structure. Then the proposed method compares the original cohort where the variants are potentially functional with the synthetic cohort where the variants are silenced. The randomization utilizes the knockoff framework that ensures that the original sequence and the synthetic sequence are "exchangeable". That is, if one replaces any part of the original sequence with its synthetic, silenced sequence, the joint distribution of genetic variants (the linkage

disequilibrium structure etc.) remains the same. This leads to an important feature of our proposed screening procedure that is similar to real functional experiments, namely the ability to prioritize causal variants over associations due to linkage disequilibrium and other unadjusted confounding effects (e.g., shadow effects of nearby significant variants and unadjusted population stratification) as we show below.

In this paper, we present a statistical approach that addresses the challenges described above, and leads to increased power to detect and localize common and rare risk variants at genome-wide scale. The framework appropriately accounts for arbitrary correlations while guaranteeing FDR control at the desired level, and therefore has higher power than existing association tests that control FWER. Furthermore, the proposed method has additional important advantages over the standard approaches due to some intrinsic properties of the underlying framework. Specifically, it allows for the prioritization of causal variants over associations due to linkage disequilibrium. For analyses specifically focusing on rare variants, the method naturally distinguishes the signal due to rare variants from shadow effects of nearby significant (common or rare) variants. Additionally, it naturally reduces false positives due to unadjusted population stratification.

## Results

**Overview of the screening procedure with multiple knockoffs (KnockoffScreen).** We describe here the main ideas behind our method, KnockoffScreen. We assume a study population of $n$ subjects, with $Y_i$ being the quantitative/dichotomous outcome value; $\mathbf{X_i} = \left(X_{i1}, \ldots, X_{id}\right)^T$ being the $d$ covariates which can include age, gender, principal components of genetic variation etc.; $\left\{G_{ij}\right\}_{1 \leq j \leq p}$ being the $p$ genetic variants in the genome. For each target window $\Phi_{kl} = \left\{j : k \leq j \leq l\right\}$, we are interested in determining whether $\Phi_{kl}$ contains any variants associated with the outcome of interest while adjusting for covariates.

The idea of the proposed method is to augment the original cohort with a synthetic cohort with genetic variants, $\left\{\widetilde{G}_{ij}\right\}_{1 \leq j \leq p}$, referred to as knockoff features. $\left\{\widetilde{G}_{ij}\right\}_{1 \leq j \leq p}$ are generated by a data driven algorithm such that they are exchangeable with $\left\{G_{ij}\right\}_{1 \leq j \leq p}$, yet they do not directly affect $Y_i$ (i.e., are "silenced", and therefore not causal). More precisely, $\left\{\widetilde{G}_{ij}\right\}_{1 \leq j \leq p}$ is independent of $Y_i$ conditional on $\left\{G_{ij}\right\}_{1 \leq j \leq p}$. Note that the knockoff generation procedure is different from the well-known permutation procedure which generates control features by permuting the samples; for such a permutation procedure, the exchangeability property between the original genetic variants and the synthetic ones does not hold and hence the FDR control cannot be guaranteed[13,14].

The screening procedure examines every target window $\Phi_{kl}$ in the genome and performs hypothesis testing in both the original cohort and the synthetic cohort, to test for association of $G_{\Phi_{kl}}$ and $\widetilde{G}_{\Phi_{kl}}$ with $Y$ respectively. As explained below, the knockoff procedure is amenable to any form of association test within the window. Let $p_{\Phi_{kl}}, \widetilde{p}_{\Phi_{kl}}$ be the resulting $p$-values. We define a feature statistic as

$$W_{\Phi_{kl}} = T_{\Phi_{kl}} - \widetilde{T}_{\Phi_{kl}}, \qquad (1)$$

where $T_{\Phi_{kl}} = -\log_{10} p_{\Phi_{kl}}$ and $\widetilde{T}_{\Phi_{kl}} = -\log_{10} \widetilde{p}_{\Phi_{kl}}$. Essentially, the observed $p$-value for each window is compared to its control

counterpart in the synthetic cohort. A threshold $\tau$ for $W_{\Phi_{kl}}$ can be determined by the knockoff filter so that the FDR is controlled at the nominal level. We select all windows with $W_{\Phi_{kl}} \geq \tau$. We additionally derived the corresponding Q-value for a window, $q_{\Phi_{kl}}$, that unifies the feature statistic $W_{\Phi_{kl}}$ and the threshold $\tau$. More details are given in the Methods section.

The knockoff construction ensures exchangeability of features, namely that $\left\{G_{ij}\right\}_{1 \leq j \leq p}$ and $\left\{\widetilde{G}_{ij}\right\}_{1 \leq j \leq p}$ are exchangeable. Hence if one swaps any subset of variants with their synthetic counterpart, the joint distribution remains the same. For instance, suppose that $G_{i1}$ and $G_{i2}$ are two genetic variants, then the knockoff generator will generate their knockoff counterparts $\widetilde{G}_{i1}$ and $\widetilde{G}_{i2}$ such that $\left(G_{i1}, G_{i2}, \widetilde{G}_{i1}, \widetilde{G}_{i2}\right) \sim \left(G_{i1}, \widetilde{G}_{i2}, \widetilde{G}_{i1}, G_{i2}\right)$, where "$\sim$" denotes equality in distribution. More generally, for any subset $S \subset \left\{1, \ldots, p\right\}$,

$$\left(G_i, \widetilde{G}_i\right)_{\text{swap}(S)} \sim \left(G_i, \widetilde{G}_i\right), \qquad (2)$$

where $\left(G_i, \widetilde{G}_i\right)_{\text{swap}(S)}$ is obtained from $\left(G_i, \widetilde{G}_i\right)$ by swapping the variants $G_{ij}$ and $\widetilde{G}_{ij}$ for each $j \in S$. This feature exchangeability implies the exchangeability of the importance scores $T_{\Phi_{kl}}$ and $\widetilde{T}_{\Phi_{kl}}$ under the null hypothesis, i.e., $\left(T_{\Phi_{kl}}, \widetilde{T}_{\Phi_{kl}}\right) \sim \left(\widetilde{T}_{\Phi_{kl}}, T_{\Phi_{kl}}\right)$ if $\Phi_{kl}$ does not contain any causal variant. Thus $\widetilde{T}_{\Phi_{kl}}$ can be used as the negative control, and we reject the null when $W_{\Phi_{kl}} = T_{\Phi_{kl}} - \widetilde{T}_{\Phi_{kl}}$ is sufficiently large. This exchangeability property leads to several interesting properties of our proposed screening procedure relative to conventional association tests as mentioned in the Introduction, and which will be discussed in detail in later sections.

Once the knockoff generation is completed, we apply a genome-wide screening procedure. Our screening procedure considers windows with different sizes (1 bp, 1 kb, 5 kb, 10 kb) across the genome, with half of each window overlapping with adjacent windows of the same size. To calculate the importance score for each window $\Phi_{kl}$, we incorporate several recent advances for association tests for sequencing studies to compute $p_{\Phi_{kl}}$.

- For each 1 bp window (i.e., single variant): we only consider common (minor allele frequency (MAF) > 0.05) and low frequency (0.01 < MAF < 0.05) variants and compute $p_{\Phi_{kl}}$ from single variant score test. For binary traits, we implement the saddlepoint approximation for unbalanced case-control data.
- For each 1 kb/5 kb/10 kb window, we perform:

  a. Burden and dispersion tests for common and low frequency variants with Beta (MAF, 1, 25) weights, where Beta (.) is the probability density function of the beta distribution with shape parameters 1 and 25[26]. These tests aim to detect the combined effects of common and low frequency variants.

  b. Burden and dispersion tests for rare variants (MAF < 0.01 & minor allele count (MAC) >= 5) with Beta (MAF, 1, 25) weights. These tests aim to detect the combined effects of rare variants.

  c. Burden and dispersion tests for rare variants, weighted by functional annotations[23]. Current implementation includes CADD[27] and tissue/cell type specific GenoNet scores[28]. These tests aim to utilize functional annotations for improved power.

  d. Burden test for aggregation of ultra-rare variants (MAC < 5). These tests aim to aggregate effects from extremely rare variants such as singletons, doubletons etc.

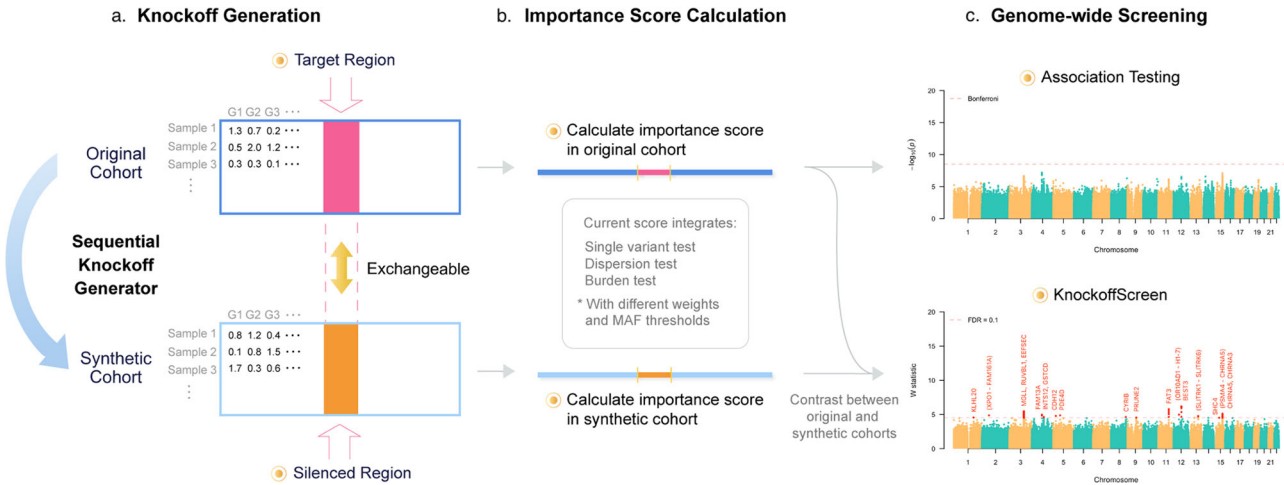

**Fig. 1 Overview of KnockoffScreen. a** Knockoff generation based on the original genotype matrix. Each row in the matrix corresponds to an individual and each column corresponds to a genetic variant. Each cell presents the genotype value/dosage. **b** Calculation of the importance score for each 1 bp, 1 kb, 5 kb, or 10 kb window. **c** Example of genome-wide screening results using conventional association testing (top) and KnockoffScreen (bottom).

e. Single variant score tests for common, low frequency and rare variants in the window.

f. The aggregated Cauchy association test[29] to compute $p_{\Phi_{kl}}$.

We also extend the single knockoff described above to the setting with multiple knockoffs to improve the power, stability and reproducibility of the findings. Let $q$ be the FDR threshold. The inference based on a single knockoff is limited by a detection threshold of $1/q$, defined as the minimum number of independent signals required for making any discovery. It has no power at the target FDR level $q$ if there are fewer than $1/q$ discoveries to be made. The multiple knockoffs improve the detection threshold from $1/q$ to $1/(Mq)$, where $M$ is the number of knockoffs[30]. For example, the detection threshold is 10 when the target FDR = 0.1. In scenarios where the signal is sparse (<10 independent causal variants) in the target region or across the genome, inference based on a single knockoff can have very low power to detect any of the causal variants. In such a setting, KnockoffScreen with $M$ knockoffs reduces the detection threshold from 10 to $10/M$, which allows KnockoffScreen to detect sparse signals in a target region or across the genome. Furthermore, integrating multiple knockoffs leads to improvements in the stability and reproducibility of the knockoff procedure. Specifically, the results of the KnockoffScreen procedure depend to some extent on the sampling of knockoff features $\left\{ \widetilde{G}_{ij} \right\}_{1 \leq j \leq p}$, which is random. Therefore, running the analysis twice on the same dataset may lead to the selection of slightly different subsets of features. In particular, for weak causal effects, there is a chance that the causal variant is selected in only one of the analyses. We demonstrate in the Methods section that our choice of multiple knockoff statistics helps improve the stability of the results compared with state-of-the-art alternatives.

In the Methods section, we describe in detail our computationally efficient method to generate the knockoff features, and our multiple knockoffs method. A flowchart of our approach is shown in Fig. 1.

**KnockoffScreen improves power and guarantees FDR control in single-region simulation studies.** We performed empirical power and FDR simulations to evaluate the performance of KnockoffScreen in a single region. We compared it with existing

alternatives for sequence-based association testing, including the burden and dispersion (SKAT) tests with Beta(MAF;1,25) weights. For a fair and simplified comparison, we did not include additional functional annotations in our method for these simulations. Note that burden and SKAT are also applied within the knockoff framework, and therefore we still aim at controlling the FDR. We also compared state-of-the-art methods for generating knockoff features, including the second-order knockoff generator proposed by Candès et al.[14], referred to as SecondOrder, and the knockoff generator for Hidden Markov Models (HMMs) proposed by Sesia et al.[15,16] with number of states S = 50. For simulating the sequence data, each replicate consists of 10,000 individuals with genetic data on 1000 genetic variants from a 200 kb region, simulated using the haplotype dataset in the SKAT package. The SKAT haplotype dataset was generated using a coalescent model (COSI), mimicking the linkage disequilibrium structure of European ancestry samples. Simulation details are provided in the Methods section. We compared the methods in different scenarios for common and rare variants, quantitative traits, and dichotomous traits. For each replicate, the empirical power is defined as the proportion of detected windows among all causal windows (windows that contain at least one causal variant); the empirical FDR is defined as the proportion of non-causal windows among all detected windows. We present the average power and FDR over 1000 replicates in Fig. 2. We additionally present the distribution of power and the false discovery proportion (FDP) at target FDR level 0.1 over 1000 replicates in Supplementary Fig. 1.

The comparisons of the different knockoff generators show that KnockoffScreen has significantly improved power with a better FDR control. For single knockoff generators, SecondOrder and HMMs have inflated FDR for rare variants. We also observed that the HMM-based knockoff has inflated FDR for common variants for the window-based screening procedure considered in this paper. KnockoffScreen has well-controlled FDR, and significantly higher power compared with a single knockoff, especially when the target FDR $q$ is small. This is due to the high detection threshold ($1/q$) needed for the single knockoff. Our multiple knockoff method KnockoffScreen incorporates five knockoffs, and as a consequence the detection threshold is reduced from $1/q$ to $1/(5q)$, which helps improve power. We note that the power of methods with single knockoff and multiple knockoffs may be comparable in settings where the detection

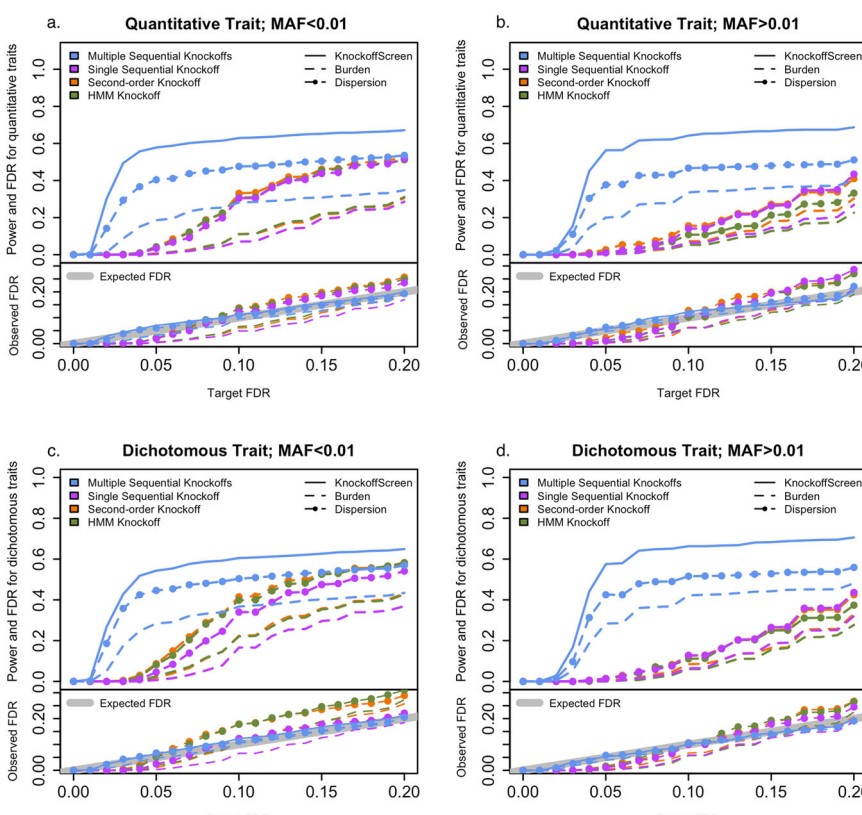

**Fig. 2 Power and false discovery rate (FDR) simulation studies in a single region.** The four panels show power and FDR base on 500 replicates for different types of traits (quantitative and dichotomous) and different types of variants (rare and common), with different target FDR varying from 0 to 0.2. The different colors indicate different knockoff generators. The different types of lines indicate different tests to define the importance score. Source data are provided as a Source Data file.

threshold is not a primary factor that limits the power, such as for higher target FDR values. Furthermore, we observed that the additional tests included in KnockoffScreen improve its power, compared to the burden and SKAT tests with the same number of knockoffs. In summary, the simulation results show that the screening procedure and multiple knockoffs help improve power while controlling FDR at the nominal level.

**KnockoffScreen improves genome-wide locus discovery for polygenic traits.** We conducted genome-wide empirical FDR and power simulations using ADSP whole-genome sequencing data to evaluate the performance of KnockoffScreen in the presence of multiple causal loci. Specifically, we randomly choose 10 causal loci and 500 noise loci across the whole genome, each of size 200 kb. Each causal locus contains a 10 kb causal window. For each replicate, we randomly set 10% variants in each 10 kb causal window to be causal. In total, there are approximately 335 causal variants on average across the genome. Simulation details are provided in the Methods section. We compared the proposed KnockoffScreen method to conventional p-value based methods including Bonferroni correction for FWER control, and BH procedure for FDR control. For KnockoffScreen we also evaluated the effect of different numbers of knockoffs. We evaluated the empirical power and FDR at target FDR 0.10. For each replicate, the power is defined as proportion of the 200 kb causal loci detected by each method; the empirical FDR is defined as the proportion of significant windows ±100/75/50 kb away from the

causal windows. We report the average power and FDR over 100 replicates in Fig. 3.

The simulation results show that KnockoffScreen exhibits substantially higher power than using Bonferroni correction. Additionally, using the conventional Benjamini-Hochberg FDR control may have higher power than KnockoffScreen, but fails to control FDR at higher resolution (e.g., ±75 kb). Statistically, the knockoff filter is expected to have similar or higher power for independent tests compared with the BH procedure[13]. For correlated genetic variants/windows, the higher empirical power of the BH procedure in our simulation studies is subject to false-positive inflation. Therefore, we do not recommend directly using the conventional BH procedure in whole genome sequencing studies. In the presence of multiple causal loci and at a moderate target FDR, we observe that the power is similar for different number of knockoffs because the aforementioned detection threshold is no longer an issue. Thus, multiple knockoffs are particularly useful when the number of causal loci is small, and the target FDR is stringent. Regardless of the effect on power, an important advantage of using multiple knockoffs is that it can significantly improve the stability and reproducibility of knockoff-based inference. Since the knockoff sampling is random, each run of the knockoff procedure may lead to different selected sets of features. In practice, strong signals will always be selected but weak signals may be missed at random with a single knockoff. The proposed multiple knockoff procedure has significantly smaller variation in feature statistic in our simulation study based on real data from ADSP. We discuss the details in the Methods section (Fig. 9).

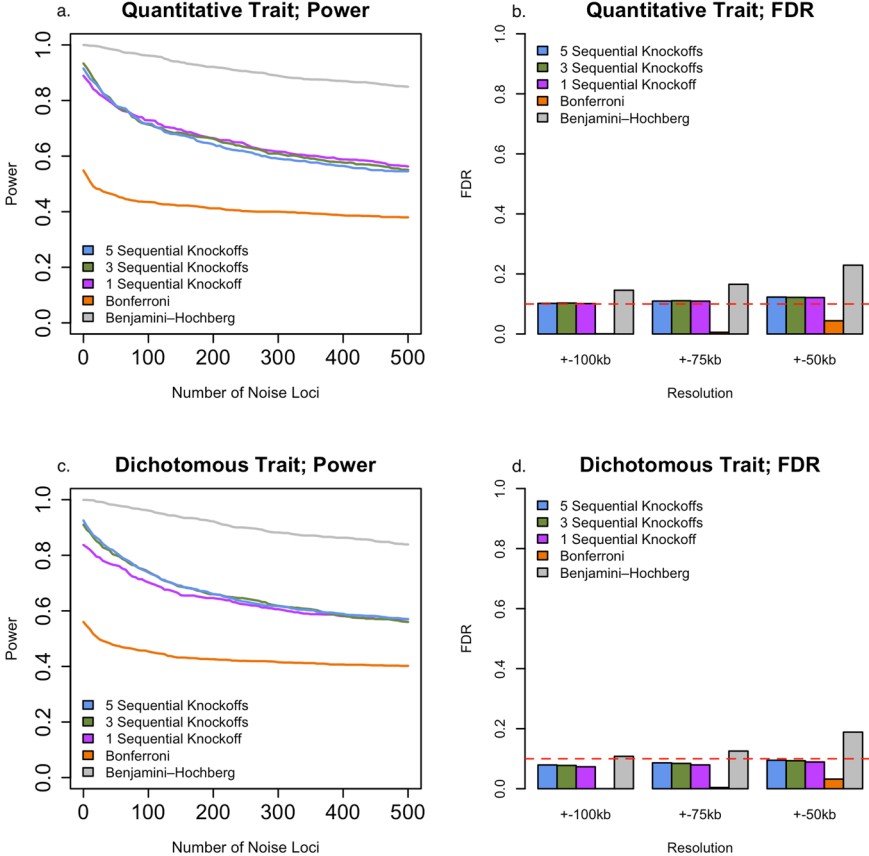

**Fig. 3 Genome-wide power and false discovery rate (FDR) simulations studies in the presence of multiple causal loci. a, c** Empirical power for different types of traits (quantitative and dichotomous), defined as the average proportion of 200 kb causal loci being identified at target FDR 0.1. **b, d** Empirical FDR for different types of traits (quantitative and dichotomous) at different resolutions, defined as the proportion of significant windows (target FDR 0.1) ± 100/75/50 kb away from the causal windows. The empirical power and FDR have averaged over 100 replicates. Source data are provided as a Source Data file.

**KnockoffScreen prioritizes causal variants/loci over associations due to linkage disequilibrium.** The exchangeability properties for the features help the inference based on the feature statistic $W_{\Phi_{kl}} = T_{\Phi_{kl}} - \widetilde{T}_{\Phi_{kl}}$ to prioritize causal variants/loci over associations due to linkage disequilibrium (LD). For example, suppose $G_{i1}$ is causal and $G_{i2}$ is a null variant correlated with $G_{i1}$; $\left(\widetilde{G}_{i1}, \widetilde{G}_{i2}\right)$ are exchangeable with $\left(G_{i1}, G_{i2}\right)$, therefore $\mathrm{cor}\left(G_{i1}, \widetilde{G}_{i2}\right) \approx \mathrm{cor}\left(G_{i1}, G_{i2}\right)$. Thus, the resulting $p$-values $p_2 \sim \widetilde{p}_2$, and hence $W_2 = -\log p_2 - \left(-\log \widetilde{p}_2\right)$ follows a distribution that is symmetric around 0. This way, by comparing the $p$-value of $G_{i2}$ (a null variant) to that of its control counterpart, the method no longer identifies the proxy variant $G_{i2}$ as significant. On the other hand, the knockoff generation minimizes the correlation between feature $G_{i1}$ and its knockoff counterpart $\widetilde{G}_{i1}$, such that $W_1 = -\log p_1 - \left(-\log \widetilde{p}_1\right)$ takes positive value with higher probability and therefore can identify the causal variant $G_{i1}$ as significant.

We compared KnockoffScreen with state-of-the-art methods which perform association tests in each window and apply a hard threshold (e.g., Bonferroni correction) to control for FWER. For a fair comparison, for the conventional association testing we adopted the same combination of tests (i.e., we combined the same single variant and region-based tests) implemented in KnockoffScreen to calculate the $p$-value. As a proof of concept, we show first the results from an analysis of common and rare variants within a 200 kb region near the apolipoprotein E (*APOE*) gene for Alzheimer's Disease (AD), using data on 3,894 individuals from

the Alzheimer's Disease Sequencing Project (ADSP). More details on the data analysis for ADSP are described in a later section. *APOE* is a major genetic determinant of AD risk, containing AD risk/protective alleles. *APOE* comes in three forms (*APOE* ε2/ε3/ε4). Among them, ε2 is the least common and confers reduced risk to AD, ε4 is the most common and increases risk to AD, while ε3 appears neutral. We found that the conventional association test using a Bonferroni correction identifies a large number of significant associations ($p < 0.05$/number of tested windows), but most of these windows are presumably false positives due to LD since they are no longer significant after adjusting for the *APOE* alleles (Fig. 4a). In contrast, KnockoffScreen filtered out a considerable number of associations that are likely due to LD, and identified more refined windows that reside in *APOE* and *APOC1* at target FDR = 0.1 (Fig. 4b). A recent study identified AD risk variants and haplotypes in the *APOC1* region, and showed that these signals are independent of the *APOE*-ε4 coding change, consistent with our findings[31].

We conducted additional simulation studies to further investigate this property. We randomly drew a subset of variants (1000 variants) from the 200 kb region near *APOE*, set a 5 kb window (similar to the size of *APOE*) as the causal window, and then simulated disease phenotypes. More details on these simulations are provided in the Methods section. With target FDR = 0.1, we evaluated the proportion of selected windows overlapping the true causal window, and the maximum distance between the selected windows and the causal window. Figure 4c, d shows the results over 500 replicates. We found that windows selected by KnockoffScreen have a significantly better chance to

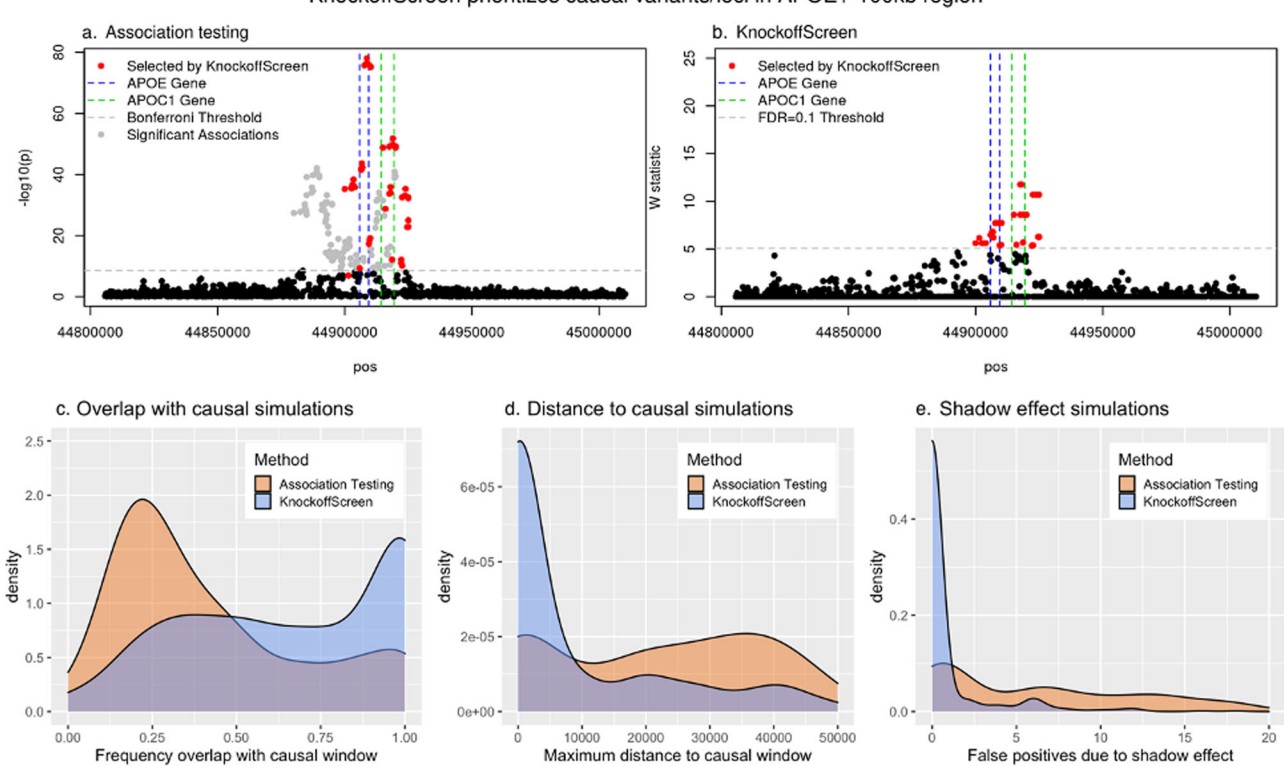

**Fig. 4 KnockoffScreen prioritizes causal variants/loci and distinguishes the signal due to rare variants from shadow effects of significant common variants nearby. a**, **b** Results of the data analyses of the APOE ± 100 kb region from the ADSP data. Each dot represents a window. Windows selected by KnockoffScreen are highlighted in red. Windows selected by conventional association testing but not by KnockoffScreen are shown in gray. **c**–**e** Simulation results based on the APOE ± 100 kb region, comparing the conventional association testing and KnockoffScreen methods in terms of **c**, frequency of selected variants/windows overlapping with the causal region; **d** Maximum distance of selected variants/windows to the causal region; **e** number of false positives due to shadow effect. The target FDR is 0.1. The density plots are based on 500 replicates. Source data are provided as a Source Data file.

overlap with the causal window relative to the conventional association test. We also found that the maximum distance between the selected windows and the causal window is significantly smaller for KnockoffScreen. Particularly, the distribution of the maximum distance to the causal window is zero-inflated for KnockoffScreen; these are cases where all windows detected by KnockoffScreen overlap/cover the causal window.

Overall, the real data example and these simulation results demonstrate that KnockoffScreen is able to prioritize causal variants over associations due to linkage disequilibrium and produces more accurate results in detecting disease risk variants/loci, thereby improving interpretation of the findings.

**KnockoffScreen distinguishes the signal due to rare variants from shadow effects of significant common variants nearby.** Conventional sequence-based association tests focused on rare variants (MAF below a certain threshold, e.g., 0.01) can lead to false positive findings by identifying rare variants that are not causal but instead correlated with a known causal common variant at the same locus; this is referred to as the shadow effects[32]. For illustration, we conducted simulation studies based on the same 200 kb region near the *APOE* gene as described above. We adopted the same simulation setting but set the causal variants to be common (MAF > 0.01) and apply the methods to rare variants only (MAF < 0.01). More details on these simulations are provided in the Methods section. Since all causal variants are common, all detected windows (focusing on rare variants) are false positives due to the shadow effect. We compared KnockoffScreen with conventional association testing by counting the number of false positives and show the distribution over 500 replicates in

Fig. 4e. For a fair comparison, for the conventional association testing we adopted the same ensemble of tests implemented in KnockoffScreen to calculate the *p*-value. We observed that the conventional tests tend to identify a large number of false positives due to the shadow effect. In contrast, KnockoffScreen has a significantly reduced number of false positives, demonstrating that it is able to distinguish the effect of rare variants from that of common variants nearby. This feature is particularly appealing in detecting novel rare association signals in whole-genome sequencing studies. The same argument also holds if instead rare variants were causal; by construction, KnockoffScreen applied to common variants only can distinguish effects attributable to common causal variants from those due to rare causal variants nearby.

**Empirical evaluation of KnockoffScreen in the presence of population stratification.** Population structure is an important confounder in genetic association studies. Standard methods to adjust for population stratification, including principal component analysis or mixed effect models, help control for global ancestry in conventional sequencing association tests. We performed an empirical evaluation of KnockoffScreen in the presence of population stratification using sequencing data from the ADSP project. We also evaluated whether, by regressing out the top principal components when computing the association statistics (*p*-values), KnockoffScreen is able to control FDR. Specifically, we randomly drew a subset of variants (1000 variants) from the 200 kb region near the *APOE* region in the ADSP study. The ADSP includes three ethnic groups: African American (AA), Non-Hispanic White (NHW), and Others (of which, 98% are

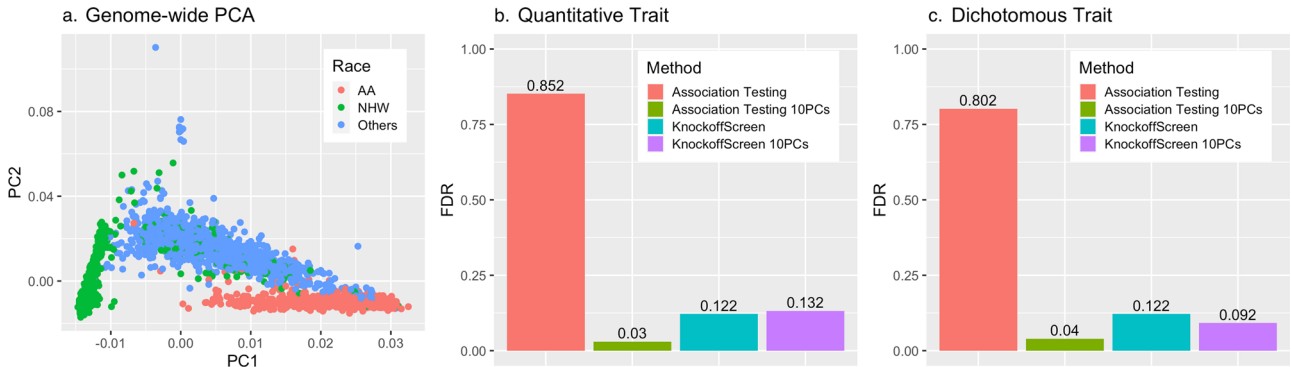

**Fig. 5 Empirical evaluation of KnockoffScreen in the presence of population stratification. a** Principal component analysis of the ADSP data, which contains three ethnic groups: African American (AA), Non-Hispanic White (NHW), and Others (of which, 98% are Caribbean Hispanic). Each dot represents an individual. **b**, **c** Simulation results for the FDR control in the presence of population stratification that mimics the ADSP data, comparing KnockoffScreen with conventional association testing. Each panel shows empirical FDR based on 500 replicates. KnockoffScreen 10PCs is a modified version of KnockoffScreen method that includes adjustment for the top principal components while computing the association statistics (*p*-values). KnockoffScreen controls FDR at 0.10; Association Testing is based on usual Bonferroni correction (0.05/number of tests), controlling FWER at 0.05. Source data are provided as a Source Data file.

Caribbean Hispanic) (see genome-wide PCA results in Fig. 5a). We set the mean/prevalence for the quantitative/dichotomous trait to be a function of the subpopulation, but not directly affected by any genetic variants. More details on these simulations are provided in the Methods section. We compared KnockoffScreen with the conventional association test with no adjustment for population stratification. We also included a modified version of KnockoffScreen that adjusts for the top 10 global PCs when computing the *p*-values used to compute the window feature statistic, referred to as KnockoffScreen+*10PCs*. For comparison, we also included the conventional association test based on Bonferroni correction, which defines significant associations by *p*-value < 0.05/number of tests.

Since in these simulations none of the genetic variants are causal, all detected windows are false positives due to the confounding effects of population structure. With a target FDR = 0.1, we calculated the observed FDR, defined as the proportion of replicates where any window is detected, and present the results in Fig. 5b, c. We observed that both PC-adjusted KnockoffScreen and the conventional PC-adjusted association test are able to control FDR at the target level. This is further illustrated by our real data analysis of ADSP where despite the combined analysis of three ethnicities there is no apparent inflation in false positive signals. Interestingly, KnockoffScreen exhibits lower FDR than association test when they are both unadjusted, indicating that the use of knockoffs naturally helps to prioritize causal variants over association due to population stratifications. We additionally performed simulation studies to mimic population stratification driven by rare variants and present the results in Supplementary Table 1. As before, we found that both PC-adjusted Knock-offScreen and association test are able to control FDR in the scenarios considered here, and KnockoffScreen exhibits a lower FDR than the conventional association test for an unadjusted model. Since the reduction of false positives for KnockoffScreen does not require observing/estimating the underlying ancestry, the knockoff procedure can potentially complement existing tools for ancestry adjustment to better reduce false positive findings due to population substructure. However, we clarify that Knock-offScreen itself does not completely eliminate the confounding due to population stratification (Supplementary Table 1) because the current knockoff generator assumes the same LD structure across individuals and it only accounts for local LD structure. Therefore, it does not capture heterogeneous LD structure across populations and strong long-range LD due to population

stratification. Development of new knockoff generators that explicitly account for population structure will be of interest[33].

**KnockoffScreen enables computationally efficient screening of whole-genome sequencing data**. One obstacle for the widespread application of knockoffs to genetic data, particularly whole-genome sequencing data, is their computational cost. The knockoff generation can be computationally intensive when the number of genetic variants *p* is large; depending on the method, it may require the calculation of the eigen values of a $p \times p$ covariance matrix, or iteratively fitting a prediction model for every variant. The whole-genome sequencing data from ADSP (~4000 individuals) contains ~85 million variants in total, much larger than the number of variants in GWAS datasets. Similarly, in 53,581 TOPMed samples, more than 400 million single-nucleotide and insertion/deletion variants were detected[2]. As more individuals are being sequenced, the number of variants will increase accordingly. We demonstrate that the proposed sequential model to simultaneously generate multiple knockoffs is significantly more computationally efficient than existing knockoff generation methods, making it scalable to whole-genome sequencing data. We compared the computing time of our proposed knockoff generator with two existing alternatives: the second-order knockoff generator proposed by Candès et al.[14], referred to as SecondOrder; and knockoffs for Hidden Markov Models (HMMs) proposed by Sesia et al.[15,16] with varying number of states (S = 12 and S = 50). We estimate the complexity of our proposed method as $O(np)$, where *n* is the sample size and *p* is the number of genetic variants. The details of this calculation are described in the Methods section. The complexity of the HMM method is also $O(np)$, as discussed in Sesia et al.[16]. However, it is significantly less efficient than the proposed method for unphased genotype data as we show below. We note that the computing time of the SecondOrder method is of order $O(np^2 + p^3)$ because it requires calculating the eigen values of a $p \times p$ covariance matrix. Therefore, it is not a feasible approach for whole-genome analysis with a large number of variants.

We performed simulations to empirically evaluate the computational time for the different methods. We note that the proposed method focuses on the analysis of whole-genome sequencing data, and thus the computational cost is reported on unphased genotype data, which is the usual format for sequencing data. Since the HMM model assumes the availability of phased

**Table 1 Computing time of different knockoff generators.**

| $n$ | $p$ | MSK (5 knockoffs) | SK | SecondOrder | HMM with S = 12 | | HMM with S = 50 | |
|---|---|---|---|---|---|---|---|---|
| | | | | | Phasing | Sampling | Phasing | Sampling |
| 1000 | 500 | 2.11 | 0.86 | 8.9 | 37.86 | 6.02 | 580.87 | 93.88 |
| 1000 | 1000 | 3.99 | 1.92 | 57.01 | 76 | 12.01 | 1147.66 | 188.74 |
| 1000 | 2000 | 8.89 | 4.06 | 491.19 | 161.94 | 24.76 | 2336.83 | 376.93 |
| 5000 | 500 | 4.66 | 1.63 | 8.51 | 188.5 | 30.45 | 2878.43 | 485.34 |
| 5000 | 1000 | 11.76 | 3.95 | 52.63 | 380.06 | 60.28 | 5914.19 | 996.11 |
| 5000 | 2000 | 31.58 | 11.09 | 479.01 | 811.61 | 129.6 | 11734.66 | 1865.11 |
| 10000 | 500 | 7.42 | 2.34 | 9.29 | 377.07 | 58.8 | 5784.24 | 957.49 |
| 10000 | 1000 | 20.57 | 6.59 | 54.66 | 757.49 | 123.94 | 11744.68 | 1936.85 |
| 10000 | 2000 | 52.86 | 16.92 | 445.05 | 1571.19 | 253.46 | 23584.8 | 3870.07 |

Each cell shows the computing time in seconds to generate knockoffs based on unphased genotype data. The multiple sequential knockoffs approach generates five knockoffs. The computing time was measured on unphased genotype data using a single CPU (Intel(R) Xeon(R) CPU E5-2640 v3 @ 2.60 GHz). Since the HMM model was mainly proposed for phased data, we report the computing time separately for phasing with fastPhase, and sampling with SNPknock.

data, we report the computing time separately for phasing with fastPhase and sampling with SNPknock as described in Sesia et al.[15]. We simulated genetic data using the SKAT package, with varying sample sizes and number of genetic variants (Table 1). The computing time was evaluated on a single CPU (Intel(R) Xeon(R) CPU E5-2640 v3 @ 2.60 GHz). For the simulation scenario considered in the previous section with 10,000 individuals and 1,000 genetic variants, we observed that the proposed method takes 6.59 s to generate a single set of knockoff features, which is ~130 times faster than the HMM model with S = 12 states (881.43 s). The application of the HMM model with the recommended S = 50 states to unphased sequencing data (13681.53 s for 10,000 individuals and 1000 genetic variants) is currently not practical at genome-wide scale. As shown, a substantial fraction of the total computing time is taken by the phasing step, and therefore using more computationally efficient phasing algorithms can further improve the computational cost of the HMM-based knockoff generation.

**KnockoffScreen detects more independent disease risk loci across the genome in two whole-genome sequencing studies.** Here we show results from the application of KnockoffScreen to two whole-genome sequencing datasets from two different studies, namely the Alzheimer's Disease Sequencing Project (ADSP), and the COPDGene study from the NHLBI Trans-Omics for Precision Medicine (TOPMed) Program. For each study, we considered windows with sizes (1 bp, 1 kb, 5 kb, 10 kb) across the genome as described before. In addition to the different weighting and thresholding strategies, we include several functional scores to improve the power of detecting rare functional variants. The functional scores include non-tissue specific CADD scores and 10 tissue/cell type specific GenoNet scores. The GenoNet scores were trained using epigenetic annotations from the Roadmap Epigenomics Project across 127 tissues/cell types. We partition the tissues/cell types into 10 groups (including Stem Cells, Blood, Connective Tissue, Brain, Internal Organs, Fetal Brain, Muscle, Fetal Tissues, and Gastrointestinal; Supplementary Table 2 has more details on these tissue groupings) and we use the maximum GenoNet score per group.

We show results from conventional association tests (using the same combination of single variant and region-based tests as implemented in KnockoffScreen) and using Bonferroni correction ($p < 0.05$/number of tested windows) to control the family-wise error rate. QQ-plots of all tests (Supplementary Fig. 5) show that the type I error rate is well controlled. We also report results from KnockoffScreen at an FDR threshold of 0.1. We assigned each significant window to its overlapping locus (gene or

intergenic region). If the locus is a gene, we report the gene's name; if the locus is intergenic, we report the upstream and downstream genes (enclosed within parentheses and separated by "-"). To assess the degree of overlap with previously described associations, we additionally searched if the loci have known associations with Alzheimer's disease and lung related traits in the NHGRI-EBI GWAS Catalog[34], acknowledging that some of the studies in the GWAS catalog included ADSP and COPDGene data. The details on gene annotations are described in the Methods section.

*Application to ADSP.* We first applied KnockoffScreen to the whole-genome sequencing data from the Alzheimer's Disease Sequencing Project (ADSP) for a genome-wide scan. The data includes 3,085 whole genomes from the ADSP Discovery Extension Study and 809 whole genomes from the Alzheimer's Disease Neuroimaging Initiative (ADNI), for a total of 3,894 whole genomes. More details on the ADSP data are provided in the Methods section. We adjusted for age, age^2, gender, ethnic group, sequencing center, and the leading 10 principal components of ancestry. We present the results in Fig. 6.

The conventional association test with Bonferroni correction identified a region (~50 kb long) at the known *APOE* locus, containing a large number of significant associations (Fig. 6), but, as discussed before, most of them are presumably due to LD with the known *APOE* risk variants since they are no longer significant after adjusting for the *APOE* alleles. Within the *APOE* region, KnockoffScreen identified fewer windows that overlap with known AD genes, namely *APOE, APOC1, APOC1P1,* and *TOMM40* at FDR < 0.1, while removing a considerable number of associations that are likely due to LD. Beyond the *APOE* locus, KnockoffScreen identified several other loci that potentially affect AD risk, including *KAT8* and an intergenic region on chromosome 18q22 between *DSEL* and *TMX3*. *KAT8* (lysine acetyltransferase 8) has been recently identified in two large scale GWAS focused on clinically diagnosed AD and AD-by-proxy individuals[35,36]. It is a promising candidate gene that affects multiple brain regions including the hippocampus and plays a putative role in neurodegeneration in both AD and Parkinson's disease[37]. The intergenic region identified by KnockoffScreen resides in a known linkage region for AD and bipolar disorder on chromosome 18q22.1[38,39]. *DSEL* (dermatan sulfate epimerase-like) is implicated in D-glucuronic acid metabolism and tumor rejection. A recent study has shown that glucuronic acid levels increase with age and predict future healthspan-related outcomes[40]. Furthermore, *DSEL* is highly expressed in the brain and has been found associated with AD in an imaging-wide

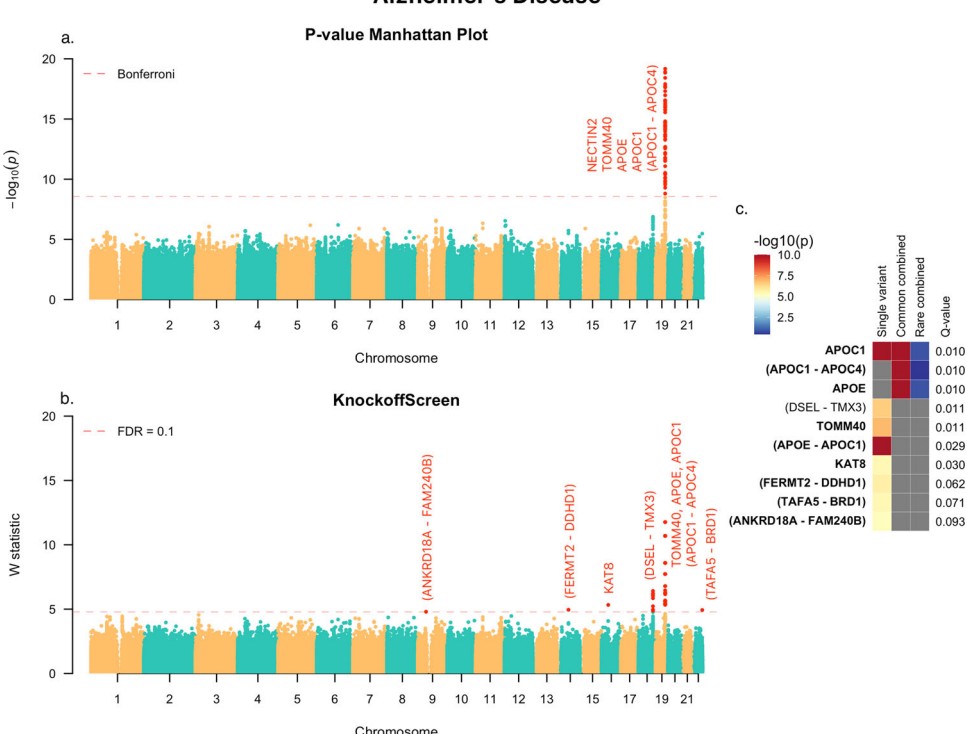

**Fig. 6 KnockoffScreen application to the Alzheimer's Disease Sequencing Project (ADSP) data to identify variants associated with the Alzheimer's Disease. a** Manhattan plot of *p*-values (truncated at $10^{-20}$ for clear visualization) from the conventional association testing with Bonferroni adjustment ($p < 0.05$/number of tested windows) for FWER control. **b** Manhattan plot of KnockoffScreen with target FDR at 0.1. **c** heatmap that shows stratified *p*-values (truncated at $10^{-10}$ for clear visualization) of all loci passing the FDR = 0.1 threshold, and the corresponding Q-values that already incorporate correction for multiple testing. The loci are shown in descending order of the knockoff statistics. For each locus, the *p*-values of the top associated single variant and/or window are shown indicating whether the signal comes from a single variant, a combined effect of common variants or a combined effect of rare variants. The names of those genes previously implicated by GWAS studies are shown in bold (names were just used to label the region and may not represent causative gene in the region). Source data are provided as a Source Data file.

association study[41]. SNPs upstream of *DSEL* have also been associated with recurrent early-onset major depressive disorder[42]. Two other intergenic loci, *ANKRD18A-FAM240B* and *TAFA5-BRD1* were reported in the GWAS catalog to have suggestive associations ($5 \times 10^{-8} < p < 1 \times 10^{-5}$) with late-onset Alzheimer's disease[43]. We additionally present results when applying the Benjamini–Hochberg procedure for FDR control in Supplementary Fig. 6; we observed that the associations identified by KnockoffScreen are largely replicated in the GWAS catalog, while the new discoveries uniquely identified using the conventional BH procedure do not overlap with previous GWAS findings, suggesting they may be false positives.

*Application to COPDGene study in TOPMed.* The Genetic Epidemiology of COPD (COPDGene) study includes current and former cigarette smokers aged >45. All subjects underwent spirometry to measure lung function. Cases were identified as those with moderate-to-severe chronic obstructive pulmonary disease (COPD), controls were those with normal lung function, and a third set were neither cases nor controls. These individuals have been whole-genome sequenced as part of the larger TOPMed project at an average ~30X coverage depth, with joint-sample variant calling and variant level quality control in TOPMed samples[2,44]. The COPDGene Freeze 5b dataset used for this analysis includes a total of 8,444 individuals, of which 5,713 are Non Hispanic White and 2,731 are African American. We tested lung function measurements on all individuals: forced expiratory volume in one second ($FEV_1$), forced vital capacity (FVC) and their ratio ($FEV_1$/FVC), as well as for case-control

COPD status on a subset (NHW: 2366 cases/2084 controls, AA: 702 cases/1409 controls).

We applied KnockoffScreen separately to the two ethnic groups, and four phenotypes, while adjusting for covariates as follows. In all analyses we adjusted for sequencing center, and the 10 leading principal components of ancestry. Additionally, for $FEV_1$ and $FEV_1$/FVC ratio, we adjusted for age, $age^2$, gender, height, $height^2$, pack-years of smoking, and current smoking. For FVC, we adjusted for age, $age^2$, gender, height, $height^2$, weight, pack-years of smoking, and current smoking. For COPD case/control status, we adjusted for age, gender, and pack-years of smoking. Results for the NHW group for $FEV_1$ are shown in Fig. 7 and those for $FEV_1$/FVC are shown in Supplementary Fig. 4.

Note that for $FEV_1$ and $FEV_1$/FVC, KnockoffScreen has been able to identify many more significant associations compared with the application to Alzheimer's disease, a reflection of the larger sample size but also the higher degree of polygenicity for lung function phenotypes relative to AD. Compared with the conventional association test with Bonferroni correction, KnockoffScreen detected several known signals for $FEV_1$, including the *PSMA4/CHRNA5/CHRNA3* locus on chromosome 15, the *INTS12/GSTCD* locus on chromosome 4, and the *EEFSEC/RUVBL1* locus on chromosome 3. Overall, the majority of the single variant signals that were found significant at FDR 0.1 have been associated with COPD-related phenotypes in the GWAS catalog (81.8% for $FEV_1$ and 69.2% for $FEV_1$/FVC) (Figs. 7 and S4) supporting the ability of KnockoffScreen to identify previously discovered loci in GWAS studies with sample sizes

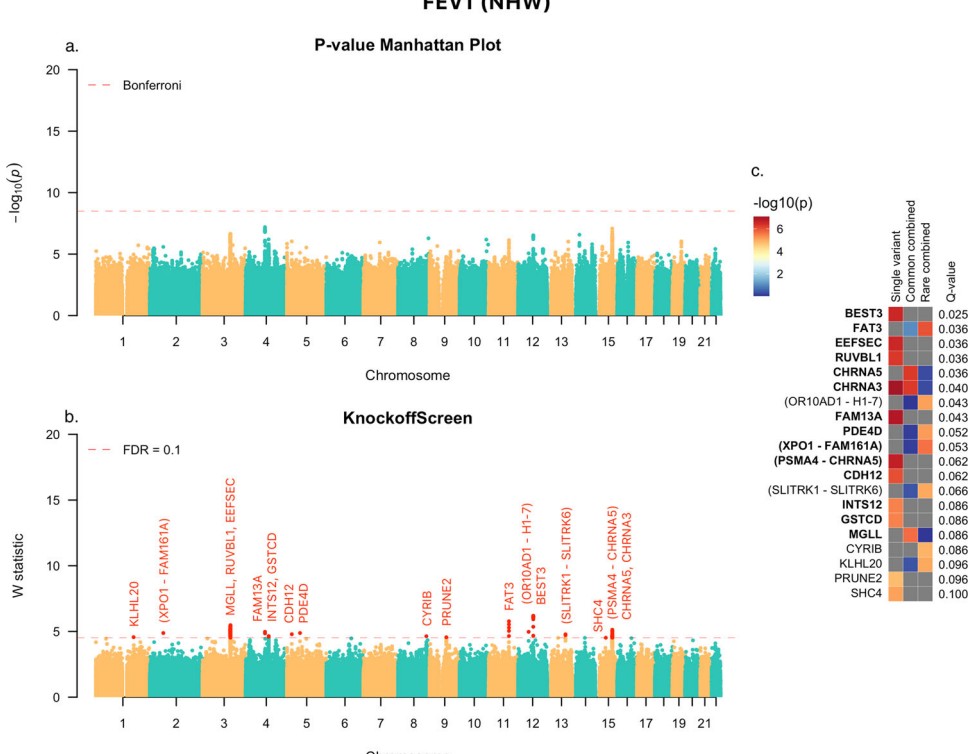

**Fig. 7 KnockoffScreen application to the COPDGene study in TOPMed to identify variants associated with FEV₁ in Non Hispanic White (NHW). a** Manhattan plot of *p*-values from the conventional association testing with Bonferroni adjustment (*p* < 0.05/number of tested windows) for FWER control. **b** Manhattan plot of KnockoffScreen with target FDR at 0.1. **c** Heatmap that shows stratified *p*-values of all loci passing the FDR = 0.1 threshold, and the corresponding Q-values that already incorporate correction for multiple testing. The loci are shown in descending order of the knockoff statistics. For each locus, the *p*-values of the top associated single variant and/or window are shown indicating whether the signal comes from a single variant, a combined effect of common variants, or a combined effect of rare variants. The names of those genes previously implicated by GWAS studies are shown in bold (names were just used to label the region and may not represent causative gene in the region). Source data are provided as a Source Data file.

much larger than used here. KnockoffScreen additionally identified new loci by aggregating common/rare variants. Although the new loci identified by KnockoffScreen, particularly those identified by rare variant methods, will need to be validated in larger datasets, and the effector genes are not known, some of the genes in these regions may be of interest. For FVC and COPD, as well as all traits for the African-Americans, we did not identify any significant associations at FDR 0.1, likely a reflection of low power due to the smaller sample size and possibly non-genetic covariates that might be associated with risk in AA and unaccounted for in these analyses.

It is interesting to note that the significant loci identified by KnockoffScreen are markedly enriched for windows (single bp or larger) overlapping protein coding genes despite an unbiased screen of the entire genome. In particular, 40%, 80%, and 56.4% of the loci significant for AD, FEV₁, and FEV₁/FVC respectively overlap protein coding genes. Given the modest sample size of the datasets analyzed here, this is perhaps expected; KnockoffScreen is able to identify the stronger effects closer to genes (e.g., coding and promoter regions). As sample sizes for whole-genome sequencing studies continue to increase, we can expect additional loci in noncoding regions to be identified.

In summary, these empirical results suggest that KnockoffScreen can identify additional signals that are missed by conventional Bonferroni correction, while filtering out proxy associations that are likely due to LD. Scatter plots comparing genome-wide W statistics vs. −log10(*p*-values) further illustrate this point (Fig. 8).

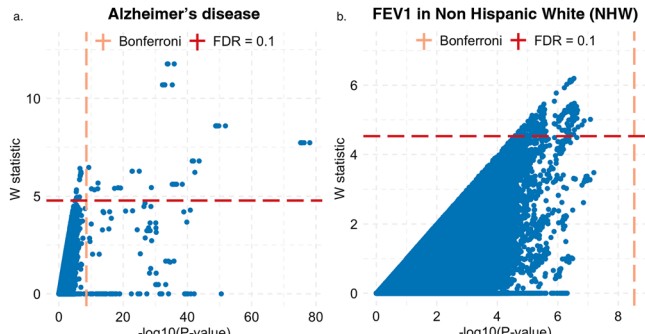

**Fig. 8 Scatter plot of genome-wide W statistic vs. −log10 (*p*-value).** Each dot represents one variant/window. The dashed lines show the significance thresholds defined by Bonferroni correction (for *p*-values) and by false discovery rate (FDR; for W statistic). The *p*-values are from the conventional association testing described in the main text. Source data are provided as a Source Data file.

## Discussion

In summary, we propose a computationally efficient algorithm, KnockoffScreen, for the identification of putative causal loci in whole-genome sequencing studies based on the knockoff framework. This framework guarantees the FDR control at the desired level under general dependence structure, and has appealing properties relative to conventional association tests, including a reduction in LD-contaminated associations and false positive

associations due to unadjusted population stratification. Through applications to two whole-genome sequencing studies for Alzheimer's disease, COPD and lung function phenotypes we demonstrate the ability of the approach to identify more significant associations, many of which have been identified in previous GWAS studies, with sample sizes orders of magnitude larger than the ones considered here. As sample sizes for whole-genome sequencing studies continue to increase, KnockoffScreen can help discover more risk loci with even more stringent FDR thresholds.

In KnockoffScreen, we choose to control FDR at the nominal level. Our analyses of data from ADSP and COPDGene show that our method compared with conventional association tests leads to significantly more discoveries. The majority of the single variant signals that were found significant at FDR 0.1 have been associated with AD or COPD-related phenotypes respectively in the GWAS catalog (87.5% for AD, 81.8% for FEV1 and 69.2% for FEV1/FVC), supporting our claim that the FDR control in KnockoffScreen is able to replicate previously discovered loci in GWAS studies with sample sizes much larger than those used here. Furthermore, KnockoffScreen identified a set of new discoveries driven by the combined effects of multiple common/rare variants. The results demonstrate that controlling FDR is an appealing strategy when there are potentially many discoveries to be made as in genetic association studies for highly polygenic traits, the dependence structure is local, and the investigators are willing to accept a rigorously defined small fraction of false positives in order to substantially increase the total number of true discoveries. We note that the choice of target FDR should be defined rigorously and interpreted appropriately. For example, loci identified at a liberal FDR threshold (e.g., 0.3 as in Iossifoy et al.9) can be useful for enrichment and pathway analyses; our analyses of data from ADSP and COPDGene used FDR = 0.1 for identifying putative causal loci. As large-scale whole-genome sequencing data become increasingly available, one will be able to apply KnockoffScreen with a lower, more stringent FDR threshold (e.g., 0.01 or 0.05).

The model-X knockoff framework underlying KnockoffScreen makes our approach robust to violations of model assumptions. Specifically, by imposing a model on genetic variants ($G_i$) instead of on the conditional distribution of the outcome given the variants (distribution of $Y_i|G_i$), the FDR control is guaranteed even when the model for $Y_i|G_i$ is mis-specified. We do however need to construct a valid synthetic cohort $\widetilde{G}_i$'s such that the exchangeability conditions are satisfied, and define a test statistic with the sign-flip property (i.e., the effect of swapping a variant with its knockoff is only a sign flip of the corresponding test statistic). This robustness feature is particularly useful for genetic studies of complex traits, as the underlying genetic model is unknown, and it is difficult to evaluate whether a model is appropriate for describing the relationship between the trait and the variants.

There is some limited work on controlling the FWER within the knockoff framework using a single knockoff[45]. One obstacle for its application is that it only allows controlling for $k$-FWER at significance level $\alpha$ (the probability of making at least $k$ false rejections) where $k$ or $\alpha$ has to be relatively large in order to detect any association. Therefore, it cannot be directly applied to control the conventional FWER ($k = 1$, $\alpha = 0.05$) without further modifications. Although our proposed multiple knockoffs method has the potential to be extended to control the FWER, we estimated that about 20 knockoffs are necessary to achieve the conventional FWER control. This leads to additional computational burden that will need to be overcome in order to become scalable to the large-scale genetic data.

In addition to controlling FDR, our approach contrasts to conventional association testing methods in that it naturally helps prioritize the underlying causal variants, a property that usually requires a second stage conditional analysis or statistical fine mapping[46]. It also helps separate causal effects from shadow effects of significant variants nearby. This property can help distinguish effects due to common causal variants or rare causal variants at the same locus due to LD, by applying KnockoffScreen to common/rare variants separately. Overall, KnockoffScreen serves as a powerful and efficient method that attempts to unify association testing and statistical fine mapping. However, similar to statistical fine-mapping methods that only leverage LD to fine-map a complex trait, it remains challenging to fully distinguish highly correlated variants. As we discussed in the Methods section, KnockoffScreen currently detects clusters of tightly linked variants, without removing any variants that are potentially causal. In the future, we may consider using functional genomics data to further improve the ability of KnockoffScreen to identify causal variants among highly correlated ones.

Unlike existing knockoff methods for genetic data that define coefficients in a LASSO regression as the importance score[15,16], KnockoffScreen directly uses transformed $p$-values as importance score. This leads to another appealing property of KnockoffScreen, namely it can serve as a wrapper method that can flexibly utilize $p$-values from any existing or future association testing methods to achieve the benefits proposed here. For example, the current implementation of KnockoffScreen calculates importance score using an ACAT type test to aggregate several recent advances for rare-variant analysis. To extend its application to studies with large unbalanced case-control ratios or sample relatedness, one can apply methods like SAIGE[47] to calculate $p$-values for the original cohort and the synthetic cohort generated by KnockoffScreen, and then apply the same knockoff filter for variable selection. Moreover, recent studies have demonstrated that multivariate models have many advantages over marginal association testing, including improved power by reducing the residual variation and better control of population stratification[15]. KnockoffScreen is able to integrate tests from multivariate models (e.g., BOLT-LMM and its extension to window-based analysis of sequencing data).

Meta-analyses are important in allowing the integration of results from multiple whole-genome sequencing studies without sharing individual level data. Several methods have been proposed for meta-analysis of single variant tests for common variants or "set" based (e.g., window based) tests for rare variants[48–50]. Those methods integrate summary statistics from each individual cohort, such as $p$-values or score statistics, and then compute a combined $p$-value for each genetic variant or each window for a meta-analysis. As we discussed, KnockoffScreen can also directly utilize $p$-values from existing methods for meta-analysis. We have discussed the detailed procedure in the Methods section.

Variable selection based on knockoff procedure depends on the random sampling of knockoff features $\left\{\widetilde{G}_{ij}\right\}_{1\leq j\leq p}$. Although FDR control is guaranteed, the randomness may lead to slightly different feature statistics and selection of slightly different subsets of variants. We propose a stable inference procedure integrating multiple knockoffs that significantly improves the stability and reproducibility of the results compared with state-of-the-art knockoff methods as discussed in the Method section.

We have demonstrated that the proposed sequential knockoff generator is significantly faster than existing alternatives. Besides the generation of knockoff features, another source of computational burden is the calculation of the importance score ($p$-value for a window). The total CPU time is 7,616 h for the ADSP data analysis (15.2 h with 500 cores) and 14,274 h for the COPDGene data analysis (28.5 h with 500 cores). The calculation of $p$-values

in the current analysis is time consuming because of the comprehensive inclusion of many different functional annotations. Specifically, for each window, there are in total 29 tests being implemented for the original genetic variants and each of their five knockoffs, leading to a total of $29*6 = 174$ $p$-value calculations per window. If computational resources are limited, using a limited number of functional annotations can substantially reduce the computing time. In addition, several methods have been proposed in recent years to use state-of-the-art optimization strategies for scalable association testing for large scale datasets with thousands of phenotypes in large biobanks.[51–53] By directly utilizing $p$-values from those association testing methods, KnockoffScreen can scale up to biobank sized datasets at a comparable computational efficiency.

Despite the aforementioned advantages, KnockoffScreen has some limitations related to underlying modeling assumptions needed to improve the computational efficiency of the multiple knockoff generation and calculation of the feature importance scores. In particular, the implemented feature importance scores rely on computing $p$-values from a marginal model (e.g., single variant score test, burden test or SKAT) or a partly multivariate model (BOLT-LMM and its extension to window-based analysis of sequencing data). We made this choice of feature importance score due to its flexibility to integrate state-of-the-art tests for sequencing studies, but we recognize that a fully multivariate model as implemented in Sesia et al.[15] can be more powerful. In addition, the knockoff generator used in KnockoffScreen assumes a linear approximation model based on unphased genotype dosage data. This model is well motivated based on the sequential model to generate knockoff features, and the approximate multivariate normal model for the genotype data commonly used in the genetic literature. Additionally, it is computationally efficient relative to existing knockoff generation methods. We acknowledge that relative to a generative model like HMM it is less interpretable. More complex models for discrete genotype values that can also account for non-linear effects among genetic variants could be of interest in future work.

## Methods

**Sequential model to generate model-X knockoff features**. We propose a computationally efficient sequential model to generate knockoff features $\widetilde{G}$ that leverages local linkage disequilibrium structure. Our method is an extension of the general sequential conditional independent pairs (SCIP) approach in Candès et al. (2018)[14].

> **Algorithm 1** Sequential Conditional Independent Pairs (Single Knockoff)
> $j = 1$
> while $j \leq p$ do
> Sample $\widetilde{G}_j$ independently from $\mathcal{L}\left(G_j | \boldsymbol{G}_{-j}, \widetilde{\boldsymbol{G}}_{1:(j-1)}\right)$
> $j = j + 1$
> **end**

where $\boldsymbol{G}_{-j}$ denotes all genetic variants except for the $j$-th variant; $\mathcal{L}\left(G_j | \boldsymbol{G}_{-j}, \widetilde{\boldsymbol{G}}_{1:(j-1)}\right)$ is the conditional distribution of $G_j$ given $\boldsymbol{G}_{-j}$ and $\widetilde{\boldsymbol{G}}_{1:(j-1)}$. Candès et al. showed that knockoffs generated by this algorithm satisfy the exchangeability condition, and they lead to a guaranteed FDR control[14]. Intuitively, the exchangeability condition can be described as follows: if one swaps any subset of variants and their synthetic counterpart, the joint distribution (LD structure etc.) does not change. They also noted that the ordering in which knockoffs are created does not affect the exchangeability property and equally valid constructions may be obtained by looping through an arbitrary ordering of the variants. Although the SCIP method represents a general knockoff generator, the conditional distribution at each iteration depends on all genetic variants in the study, which can be very difficult or impossible to compute in practice. We draw inspiration from Markov models for sequence data to consider the genetic sequence as a Markov chain with memory, such that

$$\mathcal{L}\left(G_j | \boldsymbol{G}_{-j}\right) = \mathcal{L}\left(G_j | \boldsymbol{G}_{k \in B_j}\right), \quad (3)$$

where the index set $B_j$ defines a subset of genetic variants "near" the $j$-th variant, which we will define later. Furthermore, by noting that the correlation among genetic variables approximately exhibits a block diagonal structure[54], under certain

model assumptions which will be specified in the Appendix, we have

$$\mathcal{L}\left(G_j | \boldsymbol{G}_{-j}, \widetilde{\boldsymbol{G}}_{1:j-1}\right) = \mathcal{L}\left(G_j | \boldsymbol{G}_{k \in B_j}, \widetilde{\boldsymbol{G}}_{1 \leq k \leq j-1, k \in B_j}\right). \quad (4)$$

To generate knockoff features from $\mathcal{L}\left(G_j | \boldsymbol{G}_{k \in B_j}, \widetilde{\boldsymbol{G}}_{1 \leq k \leq j-1, k \in B_j}\right)$, we assume a semiparametric model

$$G_j = g\left(\boldsymbol{G}_{k \in B_j}, \widetilde{\boldsymbol{G}}_{1 \leq k \leq j-1, k \in B_j}\right) + \varepsilon_j, \quad (5)$$

where $\varepsilon_j$ is a random error term, $E\left(\varepsilon_j | \boldsymbol{G}_{k \in B_j}, \widetilde{\boldsymbol{G}}_{1 \leq k \leq j-1, k \in B_j}\right) = 0$. We consider $g(\bullet)$ to be parametric as follows,

$$g\left(G_{ij} | \boldsymbol{G}_{k \in B_j}, \widetilde{\boldsymbol{G}}_{1 \leq k \leq j-1, k \in B_j}\right) = \alpha + \sum_{k \neq j, k \in B_j} \beta_k G_{ik} + \sum_{k \leq j-1, k \in B_j} \gamma_k \widetilde{G}_{ik}, \quad (6)$$

and will explain in detail when such a linear form is an appropriate model in the Appendix. We estimate $(\alpha, \boldsymbol{\beta}, \boldsymbol{\gamma})$ by minimizing the mean squared loss. Let $\hat{G}_j = \hat{\alpha} + \sum_{k \neq j, k \in B_j} \hat{\beta}_k \boldsymbol{G}_k + \sum_{k \leq j-1, k \in B_j} \hat{\gamma}_k \widetilde{\boldsymbol{G}}_k$. We calculate the residual $\hat{\varepsilon}_j = \boldsymbol{G}_j - \hat{\boldsymbol{G}}_j$ and its permutation $\hat{\varepsilon}_j$, and then define the knockoff feature for $\boldsymbol{G}_j$ to be $\widetilde{\boldsymbol{G}}_j = \hat{\boldsymbol{G}}_j + \hat{\varepsilon}_j$. This permutation-based algorithm is particularly designed to generate knockoff features for rare genetic variants in sequencing studies, whose distribution is highly skewed and zero-inflated. We note that the algorithm does not generate categorical variables in $\{0, 1, 2\}$. Instead, it generates continuous variables to mimic genotype dosage value, making it more robust for rare variants. In addition, we evaluated a multinomial logistic regression model for generating categorical knockoffs. We found that the conditional mean of a rare variant can be extremely small, and it is very likely to generate knockoffs with all 0 values where statistical inference cannot be applied. We show in simulation studies that existing knockoff generators, such as the second-order model-X knockoffs proposed by Candès et al.[14] and knockoffs for HMM proposed by Sesia et al.[15,16], do not control FDR for rare variant analysis based on the feature score considered in this paper (Fig. 2). In Figs. S7 and S8, we present an additional comparison between the proposed method and HMM-based knockoff generators (S = 12 and S = 50), stratified by allele frequency. As shown, the proposed method generates knockoff versions for rare variants with better exchangeability with the original variants compared with the HMM model. That is, the correlation coefficients are closer to those for the original variants for KnockoffScreen compared to HMM (bottom panel, the dots are mostly above the diagonal line). One plausible explanation is that the application of HMM to whole genome sequencing data requires accurate phased data for rare variants, which itself is a challenging task and also an active research area.

We discuss now in detail how we define $B_j$ while taking into account the linkage disequilibrium (LD) structure in the neighborhood of $j$. Let $r_{jk}$ be the sample correlation coefficient between variants $j$ and $k$. We define $B_j$ to include "$K$-nearest" genetic variants within a 200 kb window ($\pm 100$ kb from the target variant)[55] using $|r_{jk}|$ as a similarity measure. The choice of the window size aims to balance accurate modeling of local LD structure and computational efficiency. The choice of $K$ is to ensure that $P\left(G_j | \boldsymbol{G}_{k \in B_j}, \widetilde{\boldsymbol{G}}_{1 \leq k \leq j-1, k \in B_j}\right)$ accurately mimics the joint distribution $P\left(G_j | \boldsymbol{G}_{-j}, \widetilde{\boldsymbol{G}}_{1:j-1}\right)$ and to avoid overfitting. We adopt the theoretical result for regression analysis with diverging number of covariates and choose to include top $K$ variants with $|r_{jk}| > 0.05$ up to $K = n^{1/3}$, which ensures that the coefficient estimations achieve asymptotic normality[56].

We note that the sequential model is flexible enough and we could consider other supervised learning techniques like Lasso, support vector regression and artificial neural networks. However, since the auto-regressive model is fitted iteratively for every variant in the genome, these methods require cross-validation at each variant level which is computationally not applicable at genome-wide scale.

**Multiple sequential knockoffs to improve power and stability**. Inference based on single knockoff is limited by the detection threshold $[\frac{1}{q}]$, which is the minimum number of independent rejections needed in order to detect any association. For example, in scenarios where the signal is sparse (<10 independent true associations) in the target region or across the genome, inference based on a single knockoff has very low power to detect any association with target FDR 0.1. Another limitation of the single knockoff is its instability. Since the knockoff sample is random, running the knockoff procedure multiple times may lead to different selected sets of features. The idea of constructing multiple knockoffs was first discussed by Barber and Candès[13] and Candès et al.[14], and further studied in detail by Gimenez and Zou[30]. However, current methods are not applicable to rare variants and not scalable to whole genome sequencing data.

We extend the above SCIP based knockoff generator procedure to multiple knockoffs ($M$ is the total number of knockoffs), as follows.

> **Algorithm 2** Sequential Conditional Independent Tuples (Multiple Knockoffs)
> $j = 1$
> while $j \leq p$ do
> Sample $\widetilde{G}_j^1, \cdots, \widetilde{G}_j^M$ independently from $\mathcal{L}\left(G_j | \boldsymbol{G}_{-j}, \widetilde{\boldsymbol{G}}_{1:j-1}^I, .., \widetilde{\boldsymbol{G}}_{1:j-1}^M\right)$
> $j = j + 1$
> **End**

Gimenez and Zou[30] proposed this general algorithm and proved that the knockoffs generated by this algorithm satisfy the extended exchangeability condition (see Appendix for precise definition and proof). Based on this general algorithm, we extend our previous sequential model to this setting to estimate $\hat{G}_j = \hat{\alpha} + \sum_{k \neq j, k \in B_j} \hat{\beta}_k G_k + \sum_{1 \leq m \leq M} \sum_{k \leq j-1, k \in B_j} \hat{\gamma}_k^m \widetilde{G}_k^m$. We calculate the residual $\hat{\varepsilon}_j = G_j - \hat{G}_j$ and its $M$ permutations $\hat{\varepsilon}_j^1, ., \hat{\varepsilon}_j^M$, and then define the knockoff feature for $G_j$ to be $\widetilde{G}_j^m = \hat{G}_j + \hat{\varepsilon}_j^{*m}$.

*Knockoff filter to define the threshold $\tau$ for FDR control.* For single knockoff, we follow the result derived by Candès et al.[14] to define the feature statistic as $W_{\Phi_{kl}} = T_{\Phi_{kl}} - \widetilde{T}_{\Phi_{kl}}$ where $T_{\Phi_{kl}} = -\log_{10} p_{\Phi_{kl}}$ and $\widetilde{T}_{\Phi_{kl}} = -\log_{10} \widetilde{p}_{\Phi_{kl}}$ and

$$\tau = \min\left\{ t > 0 : \frac{1 + \#\left\{ \Phi_{kl} : W_{\Phi_{kl}} \leq -t \right\}}{\#\left\{ \Phi_{kl} : W_{\Phi_{kl}} \geq t \right\}} \leq q \right\}, \quad (7)$$

where "#" denote the number of elements in the set; $q$ is the target FDR level. We select all windows with $W_{\Phi_{kl}} > \tau$. For multiple knockoffs, we modify the result in Gimenez and Zou[30] and define

$$W_{\Phi_{kl}} = \left( T_{\Phi_{kl}} - \underset{1 \leq m \leq M}{\text{median}} T_{\Phi_{kl}}^m \right) I_{T_{\Phi_{kl}} \geq \max_{1 \leq m \leq M} T_{\Phi_{kl}}^m}, \quad (8)$$

and

$$\tau = \min\left\{ t > 0 : \frac{\frac{1}{M} + \frac{1}{M} \#\left\{ \Phi_{kl} : \kappa_{\Phi_{kl}} \geq 1, \tau_{\Phi_{kl}} \geq t \right\}}{\#\left\{ \Phi_{kl} : \kappa_{\Phi_{kl}} = 0, \tau_{\Phi_{kl}} \geq t \right\}} \leq q \right\}, \quad (9)$$

where $T_{\Phi_{kl}}^m = -\log p_{\Phi_{kl}}^m$; $I_{\bullet}$ is an indicator function, $I_{T_{\Phi_{kl}} \geq \max_{1 \leq m \leq M} T_{\Phi_{kl}}^m} = 1$ if $T_{\Phi_{kl}} \geq \max_{1 \leq m \leq M} T_{\Phi_{kl}}^m$ and 0 otherwise; $\kappa_{\Phi_{kl}} = \text{argmax}_{0 \leq m \leq M} T_{\Phi_{kl}}^m$ denote the index of the original (denoted as 0) or knockoff feature that has the largest importance score; $\tau_{\Phi_{kl}} = T_{\Phi_{kl}}^{(0)} - \text{median}_{1 \leq m \leq M} T_{\Phi_{kl}}^{(m)}$ denote the difference between the largest importance score and the median of the remaining importance scores. It reduces to the knockoff filter for single knockoff when $M = 1$. Essentially, $W_{\Phi_{kl}} > \tau$ selects windows where the original feature has higher importance score than any of the $M$ knockoffs (i.e., $\kappa_{\Phi_{kl}} = 0$), and the gap with the median of knockoff importance score is above some threshold.

We note that this definition of feature statistic and knockoff filter is a modified version of that proposed by Gimenez and Zou[30], where they considered the maximum instead of the median of the knockoff importance scores, i.e., $\kappa_{\Phi_{kl}} = \text{argmax}_{0 \leq m \leq M} T_{\Phi_{kl}}^m$, $\tau_{\Phi_{kl}} = T_{\Phi_{kl}}^{(0)} - \max_{1 \leq m \leq M} T_{\Phi_{kl}}^m$ and $W_{\Phi_{kl}} = \left( T_{\Phi_{kl}} - \max_{1 \leq m \leq M} T_{\Phi_{kl}}^m \right) I_{T_{\Phi_{kl}} \geq \max_{1 \leq m \leq M} T_{\Phi_{kl}}^m}$. To improve stability and reproducibility of knockoff based inference, we change $\tau_{\Phi_{kl}}$ from $T_{\Phi_{kl}}^{(0)} - \max_{1 \leq m \leq M} T_{\Phi_{kl}}^{(m)}$ to $T_{\Phi_{kl}}^{(0)} - \text{median}_{1 \leq m \leq M} T_{\Phi_{kl}}^{(m)}$. The modified method reduces the randomness coming from sampling knockoff features given the fact that sample median has much smaller variation than each individual sample or the sample maximum.

*Knockoff Q-value.* The Q-value in statistics is similar to the well-known *p*-value, except that it measures significance in terms of the FDR[57] rather than the FWER and already incorporates correction for multiple testing. For multiple hypothesis testing, a general mathematical definition of the Q-value for a null hypothesis is the minimum FDR that can be attained when all tests showing evidence against the null hypothesis at least as strong as the current one are declared as significant[58]. For example, the Q-value for usual FDR control based on ordered *p*-values can be estimated by,

$$q = \min_{t \geq p} \widehat{FDR}(t), \quad (10)$$

where $p$ is the p-value of the hypothesis under consideration and $\widehat{FDR}(t)$ is the estimated FDR if we are to reject all tests with *p*-values less than $t$. In order to introduce a more informative and interpretable measure of significance for the top signals, we extend the Q-value framework for the usual FDR control to the knockoffs based case. The proposed Q-value combines the information from both feature importance statistics $W_{\Phi_{kl}}$ and the threshold $\tau$. It also makes results comparable even we choose different feature importance statistics across multiple runs. By definition, we shall see that selecting windows with $q_\Phi < q$, where $q$ is the target FDR, is equivalent to the aforementioned knockoff filter which selects those with $W_\Phi > \tau$.

For single knockoff, we define the Q-value for window $\Phi$ with feature statistic $W_\Phi > 0$ as,

$$q_\Phi = \min_{t \leq W_\Phi} \frac{1 + \#\left\{ \Phi_{kl} : W_{\Phi_{kl}} \leq -t \right\}}{\#\left\{ \Phi_{kl} : W_{\Phi_{kl}} \geq t \right\}}, \quad (11)$$

where $\frac{1 + \#\left\{ \Phi_{kl} : W_{\Phi_{kl}} \leq -t \right\}}{\#\left\{ \Phi_{kl} : W_{\Phi_{kl}} \geq t \right\}}$ is an estimate of the proportion of false discoveries if we are to select all windows with feature statistic greater than $t > 0$, referred to as the knockoff estimate of FDR[13]. For window $\Phi$ with feature statistic $W_\Phi \leq 0$, we define $q_\Phi = 1$ and the window will never be selected. For multiple knockoffs, we define the Q-value for window $\Phi$ with statistics $\kappa_\Phi = 0$ and $\tau_\Phi$ as

$$q_\Phi = \min_{t \leq \tau_\Phi} \frac{\frac{1}{M} + \frac{1}{M} \#\left\{ \Phi_{kl} : \kappa_{\Phi_{kl}} \geq 1, \tau_{\Phi_{kl}} \geq t \right\}}{\#\left\{ \Phi_{kl} : \kappa_{\Phi_{kl}} = 0, \tau_{\Phi_{kl}} \geq t \right\}}, \quad (12)$$

where $\frac{\frac{1}{M} + \frac{1}{M} \#\left\{ \Phi_{kl} : \kappa_{\Phi_{kl}} \geq 1, \tau_{\Phi_{kl}} \geq t \right\}}{\#\left\{ \Phi_{kl} : \kappa_{\Phi_{kl}} = 0, \tau_{\Phi_{kl}} \geq t \right\}}$ is an estimate of the proportion of false discoveries if we are to select all windows with feature statistic $\kappa_{\Phi_{kl}} = 0, \tau_{\Phi_{kl}} \geq t$, which is our extension of the knockoff estimate of FDR to multiple knockoffs. For window $\Phi$ with $\kappa_\Phi \neq 0$, we again define $q_\Phi = 1$ and the window will never be selected.

**Choice of windows for genome-wide screening.** KnockoffScreen considers windows with different sizes (1 bp, 1 kb, 5 kb, 10 kb) across the genome, with half of each window overlapping with adjacent windows at the same window size. This choice of windows is similar to the scan statistic framework, WGScan, for whole-genome sequencing data[18]. It is also similar to that in *KnockoffZoom* proposed by Sesia et al.[15] for GWAS data where they also consider windows of different sizes; for each fixed window size the windows are non-overlapping but smaller windows are fully nested within larger windows. We theoretically prove the FDR control using the proposed statistic in the Appendix for nonoverlapping windows; however, the theoretical justification for the more general setting of overlapping windows remains an open question. For the proposed choice of overlapping windows, we demonstrate via empirical simulation studies that the FDR is well controlled (Fig. 2) as window overlapping is a local phenomenon.

**KnockoffScreen improves stability and reproducibility of knockoff-based inference.** We conducted simulation studies to compare KnockoffScreen with single knockoff approach, and the multiple knockoffs approach proposed by Gimenez and Zou, referred to as MK-Maximum[30].

We designed these simulations to mimic the real data analysis of ADSP. For each replicate, we randomly drew 1,000 variants, including both common and rare variants, from the 200 kb region near gene *APOE* (chr19: 44905796–44909393). We set 1.25% variants to be causal, all within a 5 kb signal window (similar to the size of *APOE*) and then simulated a dichotomous trait as follows

$$g(\mu_i) = \beta_0 + X_{i1} + \beta_1 g_1 + \ldots + \beta_s g_s,$$

where $g(x) = \log(\frac{x}{1-x})$ and $\mu_i$ is the conditional mean of $Y_i$; $\beta_0$ is chosen such that the prevalence is 10%. We set the effect $\beta_j = 0.7 \left| \log_{10} m_j \right|$, where $m_j$ is the MAF for the *j*-th variant. Given the same genotype and phenotype data, we first generated 100 knockoffs. Then we repeatedly drew five knockoffs randomly among them for 100 replicates. For each replicate, we scanned the regions with candidate window sizes (1 bp, 1 kb, 5 kb, 1 kb) using KnockoffScreen, the multiple knockoffs feature statistic based on sample maximum by Gimenez and Zou[30], and the single knockoff method. For a fair comparison, we adopted the same tests implemented in KnockoffScreen to calculate the *p*-value for all comparison methods. We calculated the variation of feature statistic $W_{\Phi_{kl}}$ for each window (stability) and the frequency with which each causal window is selected (reproducibility) over 100 replicates. We present the results in Fig. 9.

In the left panel, we observed that KnockoffScreen has significantly smaller variation in feature statistic $W_{\Phi_{kl}}$ than the other two comparison methods. We note that the method based on sample maximum, MK-Maximum, exhibits comparable and sometimes even larger variation than the method based on single knockoff. In the mid panel, we observed that KnockoffScreen has a higher chance (~0.94) to replicate findings across different knockoff replicates compared to MK-Maximum (~0.74–0.83) and single knockoff (~0.43). This improvement is further demonstrated in the right panel, where we show that KnockoffScreen exhibits smaller variation in feature statistics for the causal windows, resulting in higher reproducibility. The significantly lower reproducibility rate for single knockoffs relative to MK-Maximum is presumably due to its higher detection threshold because it exhibits similar level of variation as MK-Maximum for the causal windows.

**Practical strategy for tightly linked variants.** Variants residing in short genetic regions can be in moderate to high LD. Although the knockoff method helps to prioritize causal variants over associations due to low/moderate LD, strong correlations can make it difficult or impossible to distinguish the causal genetic variants from their highly correlated variants (see also Sesia et al.[16]). In fact, the knockoff method will rank all those highly correlated variants lower, which diminishes the power if causal variants exist (see below for a concrete example). We are primarily interested in the identification of relevant clusters of tightly linked variants, rather than individual variants. To address this issue, we propose a practical solution by slightly modifying $B_j$. The resulting algorithm improves the

power to detect clusters of tightly linked variants, without removing any variants that are potentially causal.

Specifically, we create a hierarchical clustering dendrogram using $|r_{jk}|$ as a similarity measure and define clusters by $|r_{jk}| > 0.75$, such that variants from two different clusters do not have a correlation greater than 0.75. To generate the knockoff feature for the $j$-th variant, we exclude variants from $B_j$ that are in the same cluster. For example, let $G_1$, $G_2$ and $G_3$ be three genetic variants; $G_1$ and $G_2$ are tightly linked with $|r_{12}| > 0.75$. The standard knockoff procedure will generate $\widetilde{G}_1$ based on $P(G_1|G_2, G_3)$, $\widetilde{G}_2$ based on $P(G_2|G_1, G_3, \widetilde{G}_1)$. Since $G_1$ and $G_2$ are highly correlated, $\widetilde{G}_1 \approx G_1$, $\widetilde{G}_2 \approx G_2$ and there will be no power to detect $G_1$ or $G_2$ even if one of them is causal. To improve the power, our modified algorithm simultaneously generates $\widetilde{G}_1$ and $\widetilde{G}_2$ based on a joint distribution $P(G_1, G_2|G_3)$ by first estimating the conditional means and then permuting the residuals jointly. This avoids the situation of $\widetilde{G}_1$ and $\widetilde{G}_2$ being identical to $G_1$ and $G_2$ because $G_1(G_2)$ is excluded from the generation of $\widetilde{G}_2(\widetilde{G}_1)$. Thus both $G_1$ and $G_2$ can be detected as a cluster. The idea is similar to that of group-wise exchangeable knockoffs proposed by Sesia et al.[15]. We further discuss limitations and some alternative approaches in the Discussion section.

**Computational efficiency of the knockoff generator.** We estimate the computational complexity of our proposed method for each variant $j$ as $O(nL) + O(L\log L) + O(n(K + MK)^2 + (K + MK)^3) = O(n)$, where $n$ is the sample size; $L$ is a predefined constant for the length of the nearby region; $K$ is the number of variants in the defined set $B_j$, which is bounded by the predefined constant $L$; $M$ is a predetermined constant for the number of knockoffs. $O(nL)$ is for calculating the correlation between variant $j$ and variants in the nearby region; $O(L\log L)$ is for the hierarchical clustering; $O(n(K + MK)^2 + (K + MK)^3)$ is for fitting the conditional auto-regressive model. Since we iteratively generate the knockoff for every variant, we estimate the complexity of our proposed method for all variants as $O(np)$, where $p$ is the number of genetic variants. We note that the genotype matrix $G$ is sparse for rare variants. Therefore, the cost for calculation of correlation and hierarchical clustering can be drastically reduced. In addition, the approach that we proposed to define $B_j$ ensures that $K$ is relatively small and this further reduces the computational cost.

**KnockoffScreen allows meta-analysis of multiple cohorts.** Meta-analysis is a powerful approach that enables integration of multiple cohorts for a larger sample size without sharing individual level data. Several methods have been proposed for meta-analysis of single variant tests for common variants or set-based (e.g., window based) tests for rare variants[48–50]. Those methods integrate summary statistics from each individual cohort, such as $p$-values or score statistics, and then compute a combined $p$-value for each genetic variant or each window for a meta-analysis. Since KnockoffScreen directly uses $p$-value as importance score, it can flexibly incorporate the aforementioned methods for a meta-analysis. The meta-analysis procedure is described as follows:

1. Generate knockoff features for each individual cohort.
2. Calculate summary statistics within each individual cohort for original data and knockoff data.
3. Apply existing meta-analysis methods to aggregate summary statistics to compute combined $p$-values $p_{\Phi_{kl}, combined}$ and $\widetilde{p}_{\Phi_{kl}, combined}$, for original data and knockoff data respectively.
4. Define $W_{\Phi_{kl}} = T_{\Phi_{kl}} - \widetilde{T}_{\Phi_{kl}}$ where $T_{\Phi_{kl}} = -\log_{10}p_{\Phi_{kl}, combined}$ and $\widetilde{T}_{\Phi_{kl}} = -\log_{10}\widetilde{p}_{\Phi_{kl}, combined}$, and apply KnockoffScreen to select putative causal variants. It naturally extends to multiple knockoffs as described above.

**Single-region empirical power and FDR simulations.** We conducted empirical FDR and power simulations. Each replicate consists of 10,000 individuals with genetic data on 1,000 genetic variants from a 200 kb region, simulated using the SKAT package. The SKAT haplotype dataset was generated using a coalescent model (COSI), mimicking the linkage disequilibrium structure of European ancestry samples. The simulations focus on both rare and common variants with minor allele frequency (MAF) <0.01 and >0.01 respectively. It has been discussed in Sesia et al.[16] that the false discovery proportion is difficult to define if the method identifies a variant that is tightly linked with the causal variant. The analysis of sequencing data targets different test units (set-based vs. single variant-based), further complicating the FDR comparisons. We note that the simulations here focus on method comparison for locus discovery to identify relevant clusters of tightly linked variants. Therefore, we simplify the simulation design in this particular section to avoid difficulties in defining the FDR in the presence of strong correlations by keeping one representative variant from each tightly linked cluster. Specifically, we applied hierarchical clustering such that no two clusters have cross-correlations above a threshold value of 0.75 and then randomly choose one representative variant from each cluster to be included in the simulation study.

We set 0.5% variants in the 200 kb region to be causal, all within a 10 kb signal window. Then we generated the quantitative/dichotomous trait as follows:

$$\text{Quantitative trait: } Y_i = X_{i1} + \beta_1 g_1 + \dots + \beta_s g_s + \varepsilon_i,$$

$$\text{Dichotomous trait: } g(\mu_i) = \beta_0 + X_{i1} + \beta_1 g_1 + \dots + \beta_s g_s,$$

where $X_{i1} \sim N(0, 1)$, $\varepsilon_i \sim N(0, 3)$ and they are all independent; $(g_1, \dots, g_s)$ are selected risk variants; $g(x) = \log(\frac{x}{1-x})$ and $\mu_i$ is the conditional mean of $Y_i$; for dichotomous trait, $\beta_0$ is chosen such that the prevalence is 10%. We set the effect $\beta_j = \frac{a}{\sqrt{2m_j(1-m_j)}}$, where $m_j$ is the MAF for the $j$-th variant. We define $a$ such that the variance due to the risk variants, $\beta_1^2 var(g_1) + \dots + \beta_s^2 var(g_s)$, is 0.05 for the simulations focusing on common variants and 0.1 for the simulations focusing on rare variants. We scan the regions with candidate window sizes (1 bp, 1 kb, 5 kb, 10 kb), and we consider several tests including the burden test, dispersion test, and Cauchy combination test to aggregate burden, dispersion, and individual variant test results (as discussed in the main text). This combined test is the method implemented in the KnockoffScreen method. A window is considered causal if it contains at least one causal variant. For each replicate, the empirical power is defined as the proportion of detected windows among all causal windows; the empirical FDR is defined as the proportion of non-causal windows among all detected windows. We simulated 500 replicates and calculated the average empirical power and FDR.

**Genome-wide empirical power and FDR simulations in the presence of multiple causal loci.** We conducted empirical FDR and power simulations using ADSP whole genome sequencing data, and compared the proposed method with state-of-the-art tests for sequencing data analysis adjusted by Bonferroni correction and Benjamini–Hochberg procedure for FDR control. We randomly choose 10 causal loci and 500 noise loci across the genome, each spanning 200 kb. Each causal locus contains a 10 kb causal window. For each replicate, we randomly set 10% variants in each 10 kb causal window to be causal. In total, there are approximately 335 causal variants on average across the genome. We generated the quantitative/dichotomous trait as follows:

$$\text{Quantitative trait: } Y_i = X_{i1} + \sum_{k=1}^{10}\left(\beta_{k1}g_{k1} + \dots + \beta_{k,k_s}g_{k,k_s}\right) + \varepsilon_i,$$

$$\text{Dichotomous trait: } g(\mu_i) = \beta_0 + X_{i1} + \sum_{k=1}^{10}\left(\beta_{k1}g_{k1} + \dots + \beta_{k,k_s}g_{k,k_s}\right) + \varepsilon_i,$$

where $X_{i1} \sim N(0, 1)$, $\varepsilon_i \sim N(0, 3)$ and they are all independent; $(g_1, \dots, g_s)$ are selected risk variants; $g(x) = \log(\frac{x}{1-x})$ and $\mu_i$ is the conditional mean of $Y_i$; for dichotomous trait, $\beta_0$ is chosen such that the prevalence is 10%. We set the effect $\beta_{kj} = \frac{a_k}{\sqrt{2m_{kj}(1-m_{kj})}}$, where $m_{kj}$ is the MAF for the $j$-th variant in causal window $k$. We define $a_k$ such that the phenotypic variance due to the risk variants for each causal locus, $\beta_{k1}g_{k1} + \dots + \beta_{k,k_s}g_{k,k_s}$, is 1. We scan the regions with candidate window sizes (1 bp, 1 kb, 5 kb, 10 kb), and we consider several tests including the burden test, dispersion test, and Cauchy combination test to aggregate burden, dispersion, and individual variant test results (as discussed in the main text). This combined test is the method implemented in the KnockoffScreen method. For each replicate, the empirical power is defined as the proportion of causal loci (the 200 kb regions) being identified; the empirical FDR is defined as the proportion of detected windows not overlapping with the causal window ± 50 kb/75 kb/100 kb, which evaluates FDR at different resolutions. The empirical power and FDR have averaged over 100 replicates.

**Simulations for investigating various properties of the KnockoffScreen method (the prioritization of causal variants, the influence of shadow effects from common variants, and robustness to population stratification).** We design these simulations to mimic the real data analysis of ADSP. For each replicate, we randomly drew 1000 variants, including both common and rare variants, from the 200 kb region near gene *APOE* (chr19: 44905796–44909393). We scanned the regions with candidate window sizes (1 bp, 1 kb, 5 kb, 1 kb) using the conventional association test and KnockoffScreen. For a fair comparison, we adopted the same tests implemented in KnockoffScreen to calculate the $p$-value for the conventional association testing method.

*Prioritization of causal variants.* We set 0.25% variants to be causal, all within a 5 kb signal window (similar to the size of *APOE*), and then simulated a dichotomous trait by

$$g(\mu_i) = \beta_0 + X_{i1} + \beta_1 g_1 + \dots + \beta_s g_s,$$

where $g(x) = \log(\frac{x}{1-x})$ and $\mu_i$ is the conditional mean of $Y_i$; for dichotomous trait, $\beta_0$ is chosen such that the prevalence is 10%. We set the effect $\beta_j = a\left|\log_{10}m_j\right|$, where $m_j$ is the MAF for the $j$-th variant. We defined $a = 1.4$ such that the risk variant has a similar odds ratio as *APOE*-$\varepsilon$4 (~3.1) given a similar MAF (~0.137)[59,60]. For each replicate, we compared the two methods in terms of (1) the proportion of selected windows that overlaps with the causal window; and (2) the maximum distance between selected windows and the causal window.

*Shadow effect*. We adopted the same simulation setting but set the causal variants to be common (MAF > 0.01) and apply the methods to rare variants only (MAF < 0.01). Since all causal variants are common, all detected windows are false positives due to the shadow effect. We counted the number of false positives and show the distribution over 500 replicates.

*Population stratification*. The ADSP includes three ethnic groups: African American (AA), Non Hispanic White (NHW), and Others (98% of which are Caribbean Hispanic). Let $Z_i$ denote the ethnic group ($Z_i = 0$: AA; $Z_i = 1$: NHW; $Z_i = 2$: Others). We simulated quantitative and dichotomous traits by

$$\text{Quantitative trait: } Y_i = X_{i1} + Z_i + \varepsilon_i$$

$$\text{Dichotomous trait: } g(\mu_i) = \beta_0 + X_{i1} + Z_i$$

where $X_{i1} \sim N(0,1)$, $\varepsilon_i \sim N(0,3)$ and they are all independent; $g(x) = \log(\frac{x}{1-x})$ and $\mu_i$ is the conditional mean of $Y_i$; for dichotomous trait, $\beta_0$ is chosen such that the prevalence is 10%. This way, the mean/prevalence for the quantitative/dichotomous trait is a function of the subpopulation, but not directly affected by the genetic variants. We counted the number of false positives and show the distribution over 500 replicates. We also calculated an estimate of the FDR, defined as the proportion of replicates where any window is detected.

*Population stratification driven by rare variants*. We carried out additional simulation studies to simulate population stratification driven by rare variants using the ADSP data. Specifically, we randomly choose 100 regions across the whole genome but outside chromosome 19 with each region of size 200 kb. Each region contains a 10 kb causal window. We randomly set 10% rare variants (MAF < 0.01; MAC > 10) in each causal window to exhibit small effects on the trait of interest, Thus the allele frequency differences across ethnic groups will lead to different disease prevalence, reflecting a population stratification driven by rare variants. Then we evaluate the FDR for the selected 200 kb region near gene *APOE* (chr19: 44905796–44909393). Since the causal variants are independent of the target region, the confounding effect will be due to population stratification. Specifically, we generated the quantitative/dichotomous trait as follows:

$$\text{Quantitative trait: } Y_i = X_{i1} + \gamma \sum_{k=1}^{100} \left( \beta_{k1} g_{k1} + \ldots + \beta_{k,k_s} g_{k,k_s} \right) + \varepsilon_{i,}$$

$$\text{Dichotomous trait: } g(\mu_i) = \beta_0 + X_{i1} + \gamma \sum_{k=1}^{100} \left( \beta_{k1} g_{k1} + \ldots + \beta_{k,k_s} g_{k,k_s} \right) + \varepsilon_{i,}$$

where $X_{i1} \sim N(0,1)$, $\varepsilon_i \sim N(0,3)$ and they are all independent; $(g_1, \ldots, g_s)$ are selected risk variants; $g(x) = \log(\frac{x}{1-x})$ and $\mu_i$ is the conditional mean of $Y_i$; for dichotomous trait, $\beta_0$ is chosen such that the prevalence is 10%. We set the effect $\beta_{kj} = \frac{a_k}{\sqrt{2m_{kj}(1-m_{kj})}}$, where $m_{kj}$ is the MAF for the $j$-th variant in causal window $k$. We define $a_k$ such that the variance due to the risk variants for each causal locus, $\beta_{k1} g_{k1} + \ldots + \beta_{k,k_s} g_{k,k_s}$, is 0.01; we set $\gamma = 0, 0.25, 0.5, 0.75$ which quantifies the magnitude of population stratification.

**The Alzheimer's disease sequencing project**. We first applied KnockoffScreen to whole-genome sequencing (WGS) data from the Alzheimer's Disease Sequencing Project (ADSP)[61]. The data include 3,085 whole genomes from the ADSP Discovery Extension Study including 1,096 Non-Hispanic White (NHW), 977 African American (AA) descent and 1,012 Caribbean Hispanic (CH). Sequencing for these samples was conducted through three National Human Genome Research Institute (NHGRI) funded Large Scale Sequencing and Analysis Centers (LSACs): Baylor College of Medicine Human Genome Sequencing Center, the Broad Institute, the McDonnell Genome Institute at Washington University. The samples were sequenced on the Illumina HiSeq X Ten platform with 150 bp paired-end reads. Additionally, the dataset includes 809 whole genomes from the Alzheimer's Disease Neuroimaging Initiative (ADNI) with 756 NHW, 28 AA, and 25 others. The samples were sequenced on the Illumina HiSeq 2000 platform with 100 bp paired-end reads. Whole-genome sequence data on 809 ADNI subjects (cases, mild cognitive impairment, and controls) have been harmonized using the ADSP pipeline for joint analysis. The ADSP Quality Control Work Group performs QC and concordance checks into an overall ADSP VCF file.

**COPDGene from the TOPMed project**. Eligible subjects in COPDGene Study (NCT00608764, www.copdgene.org) were of non-Hispanic white (NHW) or African-American (AA) ancestry, aged 45–80 years old, with at least 10 pack-years of smoking and no diagnosed lung disease other than COPD or asthma[62]. IRB approval was obtained at all study centers, and all study participants provided written informed consent. All subjects underwent a baseline survey, including demographics, smoking history, and symptoms; pre- and post-bronchodilator lung function testing; and chest CT scans. Samples from COPDGene were sequenced at the Broad Institute and at the Northwest Genomics Center at the University of Washington. Variants for all TOPMed samples were jointly called by the Informatics Research Center at the University of Michigan. For details on sequencing and variant calling methods, see https://www.nhlbiwgs.org/topmed-whole-genome-sequencing-project-freeze-5b-phases-1-and-2. QC included comparison of annotated and genetic sex and comparison of genotypes from prior SNP array data with genotypes called from sequencing. Samples with questionable identity from either of these checks were excluded from analysis.

**Gene annotation of the identified windows**. The windows (single bp or larger) identified as significant at a target FDR threshold are mapped to genes or intergenic regions using the human genome assembly GRCh38.p13 from the Ensembl Release 99[63]. We assign each significant window to its overlapping locus (gene or intergenic region). If the locus is a gene, we report the gene's name; if the locus is intergenic, we report the upstream and downstream genes (enclosed within parentheses and separated by "-"). We also check if the assigned locus has known associations with Alzheimer's disease and lung related traits in the NHGRI-EBI GWAS Catalog[34]. Specifically, we look up associations with the following seven traits for the ADSP: Alzheimer's disease, late-onset Alzheimer's disease, family history of Alzheimer's disease, t-tau measurement, p-tau measurement, amyloid-beta measurement, and beta-amyloid 1–42 measurement; and associations with the following 20 traits for the COPDGene: FEV1/FEC ratio, FEV1, FVC, PEF (peak expiratory flow), COPD, response to bronchodilator, asthma, chronic bronchitis, lung carcinoma, lung adenocarcinoma, pulmonary artery enlargement, FEV change measurement, pulmonary function measurement, carbon monoxide exhalation measurement, airway responsiveness measurement, serum IgE measurement, smoking behavior measurement, smoking status measurement, smoking behaviour, and smoking initiation. These annotations are shown in Supplemental Tables.

**Reporting summary**. Further information on research design is available in the Nature Research Reporting Summary linked to this article.

## Data availability

The manuscript used data from existing studies from COPDGene (TopMED, dbGaP phs000951.v4.p4) and the Alzheimer's Disease Sequencing Project (dbGaP phs000572.v8.p4). Source data are provided with this paper.

## Code availability

We have implemented KnockoffScreen in a computationally efficient R package that can be applied generally to the analysis of other whole-genome sequencing studies. The package can be accessed at: https://cran.r-project.org/web/packages/KnockoffScreen/index.html.

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

## Acknowledgements

This research is supported by NIH/NIA award AG066206 (Z.H.) and NIH/NIMH awards MH106910 and MH095797 (I.I.-L.). We gratefully acknowledge the studies and participants who provided biological samples and data for ADSP and TOPMed. The full study-specific acknowledgements are detailed in the Supplementary Note.

## Author contributions

Z.H., L.L., and I.I.-L. developed the concepts for the manuscript and proposed the method. Z.H., L.L., S.M., H.T., M.G., and I.I.-L. designed the analyses and applications and discussed results. Z.H., C.W., Y.L., J.L., and F.L. conducted the analyses. E.K.S., S.G., and M.H.C. helped interpret the results of the TOPMed analyses. Z.H., L.L., and I.I.-L. prepared the manuscript and all authors contributed to editing the paper.

## Competing interests

The authors declare no competing interests.
