## [Peer Review File · Nature Communications]

REVIEWER COMMENTS

Reviewer #1 (Remarks to the Author):

This paper proposes a computationally efficient knockoff for whole genome sequencing studies. Knockoff is a recent development in statistics and can be a useful framework in genetic studies. There are several ideal properties in the proposed method. The method is much faster than the existing HMM based approaches. It uses transformed p-values, so easily incorporates existing association tests. But there are several concerns in the paper. More experiments to investigate the performance of the method is needed.

1. In the real data analysis, the authors show that the proposed approach can identify a large number of associations while conventional p-values with the Bonferroni correction fails to identify even a single association (Figure 8). It is very surprising and also puzzling, because I think the use of knockoff can reduce the association signal if LD is stretched over long range. Also, knockoff screen introduces additional random variation in G^{\sim} , so T^{\sim} , which is randomly generated (although multiple knockoffs can alleviate this to a certain degree). Is it mainly due to the use of FDR?
2. Simulation studies are done using cosi simulated data in a small region of (200kb). More realistic simulation studies (Genome-wide, using real data) are needed to be done.
3. In Figure 2 (power simulation), can you add a conventional p-value based approach with Bonferroni correction and FDR control? Conventional FDR control (such as BH) and oracle approach (used in Knockoff zoom paper) can be added. They will be very helpful to figure out where the power improvement comes from.
4. In simulation studies and real data analysis, how about the relationship between $-\log_{10} P$ and W ? Are they highly concordant or there are regions with very different values?
5. Figure 2 shows that multiple knockoffs greatly improve power, but I am wondering whether it is because that the authors considered a small area (200 kb) with one causal region. The authors should compare different methods on a genome-wide scale with many casual regions.
6. More investigation on the multiple knockoffs will be helpful. This paper considered five knockoffs. Do more knockoffs (more than 5) further increase the power? How can researchers decide the number of knockoffs?
7. Population stratification adjustment: the authors compared the proposed method with unadjusted association tests. But the authors should compare it with PC-adjusted association tests, since no one uses association tests without PS adjustment (even when testing a seemly homogeneous population). Also, the method cannot perfectly control PS as discussed by the author (cannot perfectly capture Z_i from G_i) and simulation results (Figure 4). Additionally, the method features causes difficulties in the PS adjustment. It uses a linear regression to generate G^{\sim} , so effectively assumes the same LD structure across individuals. The regression coefficients are not varying by populations. With different LD, exchangeability cannot be achieved. The author should clearly state that the method should be applied to the homogeneous population.
8. It isn't clear how PCs are adjusted in KnockoffScreen+10PCs.
9. In GWAS, it is common to have related individuals, but the method can account for sample relatedness.
10. Please report the computation time in ADSP and COPDGene data analysis.

Reviewer #3 (Remarks to the Author):

This paper describes a new method to identify putative causal loci (both rare and common variants) for a trait in whole-genome sequencing studies using recent advances in the theory of model-X knockoffs. The proposed method combines new ideas for generating genotype knockoffs that are appropriate for WGS studies with existing work on rare-variant tests and screening methods. The use of the knockoff framework brings with the advantages of a null that tests conditional independence i.e. is a locus associated with trait conditional on other loci. Thus, loci discovered using this approach are more likely to represent causal variants.

The paper is well-written with detailed and convincing experiments that demonstrate, for the most part, the utility of the proposed method over related approaches (notably, the approaches based on standard association tests and prior approaches for generating knockoffs). Overall, this work is a valuable contribution to the analysis of WGS studies. I do have a few major comments for the authors.

Major comments:

1. To show that KnockoffScreen is useful in prioritizing causal variants over variants that are in LD, the experiments should compare to methods explicitly designed for fine-mapping e.g. CAVIAR and/or SUSIE.
2. The robustness of KnockoffScreen to population stratification is interesting. Given that the method is particularly useful in the context of rare variants and that prior studies have hinted at differential confounding at rare variants that may not be adequately accounted for by principal components (Mathieson and McVean Nature Genetics 2012), testing the method under a model of differential confounding at rare vs common variants could further showcase the utility of the method.
3. As the authors acknowledge, the procedure for generating knockoffs results in continuous-values genotypes. It is unclear how tests that require categorical genotypes are adapted to this setting (e.g. how would the burden test work in this setting?)
4. I wonder what motivated the choice of the specific model for generating knockoffs. Since this model is not "exact" (as it generates continuous-valued genotypes), I presume the authors chose this for computational efficiency. Alternatives such as multinomial logistic regression model would produce categorical outcomes with additional computational costs. The authors should discuss this choice.
5. If the number of variants K is set to be $n^{1/3}$, then the runtime-per-variant would scale $O(n^{5/3})$ rather than $O(n)$.
6. Why is the target FDR different between African-Americans and non-Hispanic Whites ?

Reviewer #4 (Remarks to the Author):

Comments on:
“KnockoffScreen: A powerful method for the identification of
putative causal loci in whole-genome sequencing data via
knockoff statistics”

September 7, 2020

1 Summary

This manuscript presents a novel method for testing variable importance in whole-genome sequencing studies using knockoffs, for the purpose of identifying likely causal variants while controlling the false discovery rate. The statistical method of knockoffs [1] has been recently proposed as a solution to account for linkage disequilibrium in genetic studies [2, 3]. The key idea of that method is to augment the data with “dummy” variables that behave in many ways similarly to the real variants, but are generated in silico by the statistician, and therefore are known to be non-causal and may be used as negative controls. Unlike permutation testing, knockoffs do not break the correlation structure among the variables, which is why they can account for linkage disequilibrium and thus help identify causal variants. This paper builds upon the framework of knockoffs, which previously focused on applications to SNP-array data from genome-wide association studies, and extends the existing methodology in different ways, focusing on the analysis of data from whole-genome studies (which include many more variants). The main components in this paper are: (1) a new algorithm for generating knockoffs for whole-genome sequencing data; (2) the deployment within this context of an existing method for combining multiple knockoffs to improve the stability of the results; (3) applications to two whole-genome sequencing studies.

Overall, the manuscript is well-written and concerns a very relevant topic. In fact, it is true that new methods may be needed for the efficient analysis of whole-genome sequencing data, which are becoming increasingly available. These data raise new and interesting computational and statistical challenges compared to SNP arrays from genome-wide association studies, since they measure many more variants. In this sense, I think the authors are on the right track. It is clear that a significant amount of work was put into this project, which I believe has the potential to lead to a good publication. However, I have some doubts about important technical details of the proposed method that I would like to see addressed. I also have some objections about the accuracy of certain statements regarding earlier methods. I am aware that fully addressing my comments may require significant changes to the proposed method, but I am convinced that these changes could lead to a stronger publication.

2 Detailed feedback

2.1 Knockoff generator

This paper proposes a novel algorithm for generating knockoffs based on a modification of the Sequential Conditional Independent Pairs (SCIP) method in [1]. Unlike previous methods for generating knockoffs of genetic data, which were based first on a multivariate Gaussian approximation [1] and then on hidden Markov models [2, 3], the approach proposed in this paper assumes variants more than 200kb apart to be independent of each other, and then models linearly the dependence of nearby variables. Two motivations are provided for this new method: faster computations, which can be useful for the analysis of whole-genome data, and the ability to easily obtain multiple knockoffs for each variant, which can improve the stability of the results [4]. However, I have some concerns about this proposal.

1. **Interpretability.** As far as I can see, the proposed approach is not based on a well-defined model for the distribution of the genetic variants. I am afraid this makes it hard to understand under what assumptions is the validity of the inferences based. This approach is unlike that of much of the existing work on knockoffs [1, 2, 4, 5], which is based on well-defined models. In particular, since no model is explicitly assumed for the joint distribution of G , it is not clear under what assumptions could the exchangeability (in the knockoff sense) of G and \tilde{G} be rigorously proved. I do not find the argument presented in the manuscript to justify the correctness of the knockoff construction to be fully satisfactory, as it appears to me to be somewhat circular.
2. **Non-linear dependencies.** Linear dependencies may not be sufficient to realistically describe the distribution of genetic variants, especially when rare variants are involved. This is why the standard models adopted in the literature for population genetics or phasing/imputation are based on hidden Markov models. For example, suppose that each individual in a population must have one of the following 4 haplotypes:

G_1	G_2	G_3
1	0	1
1	1	0
0	1	1
0	0	0

Clearly, G_3 is not independent of G_1, G_2 (knowing both G_1 and G_2 can be used to predict G_3 exactly); however, it is linearly uncorrelated with both of them. The method proposed in this paper would treat G_3 as independent of G_1 and G_2 , which does not account correctly for linkage disequilibrium in this example. It is well known that the magnitude of linear correlations is affected by the allele frequency of the variants [6], so I am afraid that the proposed approach may not accurately capture linkage disequilibrium especially for rarer variants.

3. **Long-range dependencies.** Variants 200kb apart are not necessarily independent. Even in homogeneous populations, where population structure is relatively weak, linkage disequilibrium can cause spurious discoveries much beyond 200 kb from the nearest causal variant [3].

If the population is stratified or admixed, dependencies have much longer range, to the point that not even variants on different chromosomes can be said to be independent [7, 8].

4. **Rare variants.** It is not entirely clear why the proposed approach is better suited to describe the distribution of rare variants compared to hidden Markov models, which are the standard probabilistic model used by many state-of-the-art phasing methods for whole-genome sequence data [9]. Hidden Markov models can account quite naturally for rare variants, whereas linear correlations may not correctly account for LD between variants with very different allele frequencies [6].
5. **Computational cost.** In theory, the proposed knockoff generator has the same computational complexity as the existing methods based on hidden Markov models. The authors mention that their implementation of the proposed method is faster than the current implementation of the state-of-the-art method for hidden Markov model knockoffs, citing [2]. However, the state-of-the-art implementation is that of [3], which is orders of magnitude faster than that of [2], and it has already been applied to much larger data sets than those considered here.

2.2 Group-wise testing

The authors mention the need to modify the original knockoff generation algorithm to avoid having zero power in the presence of tightly linked variants. They cite the variant pruning approach in [1, 2] as the current state-of-the-art to deal with this issue, and then propose an alternative solution based on group-wise exchangeable knockoffs. However, the current state-of-the-art in dealing with knockoffs for tightly linked variants is that of [3], which proposes a solution similar to that advanced here. Despite similar ideas, the method proposed in this paper is not fully correct. The problem in the approach proposed here is that the knockoffs for G_1 and G_2 should be generated *jointly* given G_3 to ensure that the correct exchangeability holds (example on page 23); see [3]. On page 23, the authors acknowledge that their solution is not rigorous, but they incorrectly state the problem is still unsolved in theory. In truth, the problem has been solved in the work of [3], of which the authors should be aware because they cite it elsewhere in their manuscript.

2.3 Population structure

It is true that knockoffs reduce false positives due to unadjusted population stratification, as mentioned by the authors and previously discussed in [2, 3]. However, this is only true if the knockoffs are correctly exchangeable with the real variants, which depends on how they are constructed. The construction proposed here treats variants that are more than 200kb apart as independent; therefore, it does not seem to account for those long-range dependencies that characterize population structure. See [3, 7] for a more complete discussion of this issue. In light of this, I am not sure it is correct to say the proposed method can account for population structure in principle. Furthermore, this manuscript suggests using marginal association tests, which are not as robust to population structure as multivariate tests [10]. That being said, it is true that regressing out the top principal components while computing the association statistics probably mitigates these weaknesses.

2.4 Association tests

Even though the framework of knockoffs is sufficiently general to accommodate different association tests [1], this manuscript seems to advocate for the use of marginal association tests in order to localize causal variants. I find this approach a little counter-intuitive. If the goal is to distinguish between causal variants and variants that are only marginally associated with the phenotype, marginal statistics are not the most powerful option. In fact, the most effective approaches for fine-mapping are multivariate; see for example [11] and references therein. Even genome-wide testing for SNP array data is now carried out with multivariate statistics by default (linear mixed models), because these are much more powerful and robust to population structure compared to marginal testing. From a statistical perspective, it is well-known that marginal statistics are less powerful than multivariate statistics unless the phenotype is monogenic, which is not the case for the applications considered in this paper. This is why knockoffs were previously applied in combination with multivariate statistics [1–3]. It is well known that knockoffs can also be applied with univariate statistics, but such approach should be expected to be less powerful and robust [7] compared to multivariate testing. I recognize some interesting advantages of univariate statistics: namely, their lower computational cost and their ease of use in combination with multiple knockoffs. However, more work may be needed to justify the trade-off, especially since efficient implementations of more powerful multivariate statistics may be possible even for full genome sequencing data.

2.5 Numerical experiments

The simulations on page 22 are intended to mimic the real data analysis of ADSP. However, ADSP is a polygenic disease, whereas there is only one causal locus in this simulation (*“we set 1.25% variants to be causal, all within a 5kb signal window”*). I am not fully convinced by this design, for the reasons outlined below.

1. I fear that having only one causal locus may partially hide the loss in power resulting from the choice of using univariate rather than multivariate test statistics. This is an important trade-off that I feel should be discussed more carefully.
2. I am not sure the comparison with the existing knockoff methods is as fair as it could be, given that those are designed to work genome-wide for the analysis of polygenic traits. These methods are already known to be relatively powerless/unstable when all causal variants are in a single locus (they require at least 10 distinct discoveries when applied at FDR level 10%, as correctly pointed out by the authors in this manuscript), but I would expect them to be generally more powerful for polygenic traits.
3. This simulation may not a very realistic representation of an efficient analysis insofar as it does not leverage the information about the phenotype contained in farther apart loci. By contrast, it would be more efficient to use multivariate statistics, such as linear mixed models (e.g., BOLT-LMM), Bayesian multivariate regression models (e.g., SUSIE), or sparse regression models (e.g., the lasso), since these can explain a larger fraction of the variance in the phenotype.

References

- [1] E. Candès, Y. Fan, L. Janson, and J. Lv. “Panning for gold: model-X knockoffs for high-dimensional controlled variable selection”. In: *J. R. Stat. Soc. B.* 80 (2018), pp. 551–577.
- [2] M. Sesia, C. Sabatti, and E. Candès. “Gene hunting with hidden Markov model knockoffs”. In: *Biometrika* 106 (2019), pp. 1–18.
- [3] M. Sesia, E. Katsevich, S. Bates, E. Candès, and C. Sabatti. “Multi-resolution localization of causal variants across the genome”. In: *Nat. Commun.* 11.1 (2020), pp. 1–10.
- [4] J. R. Gimenez and J. Zou. “Improving the stability of the knockoff procedure: multiple simultaneous knockoffs and entropy maximization”. In: *The 22nd International Conference on Artificial Intelligence and Statistics*. PMLR, 2019, pp. 2184–2192.
- [5] S. Bates, E. Candès, L. Janson, and W. Wang. “Metropolized knockoff sampling”. In: *Journal of the American Statistical Association* (2020), pp. 1–15.
- [6] N. R. Wray. “Allele frequencies and the r^2 measure of linkage disequilibrium: impact on design and interpretation of association studies”. In: *Twin Research and Human Genetics* 8.2 (2005), pp. 87–94.
- [7] M. Sesia, S. Bates, E. Candès, J. Marchini, and C. Sabatti. “Controlling the false discovery rate in GWAS with population structure”. In: *bioRxiv* (2020).
- [8] S. Bates, M. Sesia, C. Sabatti, and E. Candès. “Causal Inference in Genetic Trio Studies”. In: *Proc. Natl. Acad. Sci. U.S.A (to appear)* (2020+).
- [9] O. Delaneau, J.-F. Zagury, M. R. Robinson, J. L. Marchini, and E. T. Dermitzakis. “Accurate, scalable and integrative haplotype estimation”. In: *Nat. Commun.* 10.1 (2019), pp. 1–10.
- [10] J. R. Klasen, E. Barbez, L. Meier, N. Meinshausen, P. Bühlmann, M. Koornneef, W. Busch, and K. Schneeberger. “A multi-marker association method for genome-wide association studies without the need for population structure correction”. In: *Nat. Commun.* 7 (2016).
- [11] G. Wang, A. K. Sarkar, P. Carbonetto, and M. Stephens. “A simple new approach to variable selection in regression, with application to genetic fine-mapping”. In: *bioRxiv* (2019), p. 501114.

We thank the reviewers for their detailed and insightful comments. In the revised version, we have made several important modifications to address these concerns. Specifically,

- (1) We have performed genome-wide simulations based on the whole-genome sequencing data from the ADSP project to evaluate the FDR control and power in comparison with standard methods for multiple testing correction such as Bonferroni adjustment and Benjamini-Hochberg (BH) procedure for FDR control.
- (2) We have also added a theoretical justification for the proposed knockoff construction assuming a simplified, but reasonable model of correlation across the genome.
- (3) We have improved the evaluation of the performance in the case of population stratification, by simulating data according to a more realistic model.
- (4) We have also included more detailed comparisons with the HMM model both in terms of computational time and performance when dealing with rare variants in sequencing studies.

Reviewer #1 (Remarks to the Author):

This paper proposes a computationally efficient knockoff for whole genome sequencing studies. Knockoff is a recent development in statistics and can be a useful framework in genetic studies. There are several ideal properties in the proposed method. The method is much faster than the existing HMM based approaches. It uses transformed p-values, so easily incorporates existing association tests. But there are several concerns in the paper. More experiments to investigate the performance of the method is needed.

Response: Thank you very much for the nice summary, and the helpful comments. Below please find our responses to your comments.

1. In the real data analysis, the authors show that the proposed approach can identify a large number of associations while conventional p-values with the Bonferroni correction fails to identify even a single association (Figure 8). It is very surprising and also puzzling, because I think the use of knockoff can reduce the association signal if LD is stretched over long range. Also, knockoff screen introduces additional random variation in $G^{\tilde{}}$, so $T^{\tilde{}}$, which is randomly generated (although multiple knockoffs can alleviate this to a certain degree). Is it mainly due to the use of FDR?

Response: Thank you for this comment. To investigate further, we have performed additional genome-wide simulations where we show that indeed the power when using Bonferroni correction is much lower than that for the *KnockoffScreen* due to its conservative control of multiple comparisons, consistent with our empirical results on the real data (Figure 3; Figure 1 in this letter). It is however important to bear in mind that the knockoff-based method is not simply based on the p-value from the association test, and that the ranking of the findings based on the knockoff-based W statistic is not the same as that based on p-values (see also the scatter plot in Figure 2 of this letter).

Figure 1: Genome-wide power and false discovery rate (FDR) simulations studies in the presence of multiple causal loci. The two left panels show power for different types of traits (quantitative and dichotomous), defined as the average proportion of

200kb causal loci being identified at target FDR 0.1. The two right panels show empirical FDR for different types of traits (quantitative and dichotomous) at different resolutions, defined as the proportion of significant windows (target FDR 0.1) +/- 100/75/50kb away from the causal windows.

2. Simulation studies are done using *cosi* simulated data in a small region of (200kb). More realistic simulation studies (Genome-wide, using real data) are needed to be done.

Response: Thank you for the suggestion. We have performed new simulation studies using genome-wide data from the ADSP project. Specifically, we randomly choose 10 causal loci and 500 noise loci across the whole genome, each of size 200kb. Each causal locus contains a 10kb causal window. For each replicate, we randomly select 10% variants in each causal window to be causal. Across the genome there are on average 335 causal variants per replicate. With this genome-wide & multiple disease loci simulation study using real data, we compared the proposed *KnockoffScreen* method to conventional p-value based methods including Bonferroni correction for FWER control, and BH procedure for FDR control. For *KnockoffScreen* we also evaluated the effect of different numbers of knockoffs. We have added a new section to present the results (Page 7, line 5).

3. In Figure 2 (power simulation), can you add a conventional p-value based approach with Bonferroni correction and FDR control? Conventional FDR control (such as BH) and oracle approach (used in *Knockoff zoom* paper) can be added. They will be very helpful to figure out where the power improvement comes from.

Response: As explained above, we have performed new simulation studies using genome-wide data from the ADSP project. We have assumed multiple disease loci, and compared with

conventional p-value based methods including Bonferroni correction for FWER control, and BH procedure for FDR control. The simulation results show that *KnockoffScreen* exhibits substantially higher power than Bonferroni correction, consistent with the real data analysis where *KnockoffScreen* is able to identify many disease-associated loci missed by Bonferroni correction. Conventional BH FDR control fails to control FDR at the target level due to the complicated correlation structure among genetic variants.

4. In simulation studies and real data analysis, how about the relationship between $-\log_{10} P$ and W ? Are they highly concordant or there are regions with very different values?

Response: As shown in simulation studies, our proposed knockoff-based method is more powerful than the p-value based methods such as Bonferroni correction, and this improvement is not only due to a more lenient control of the type-1 error (FDR vs. FWER) but also a prioritization of causal variants over associations due to LD. We have also added scatter plots (Figure 8; Figure 2 in this letter) comparing genome-wide W statistics vs. $-\log_{10}(\text{p-values})$ in real data analysis. It shows that *KnockoffScreen* identifies additional signals that are missed by Bonferroni correction, while filtering out proxy associations that are likely due to LD.

Figure 2: Scatter plot of genome-wide W statistic vs. $-\log_{10}(\text{p-value})$. Each dot represents one variant/window. The dashed lines show the significance thresholds defined by Bonferroni correction (for p-values) and by false discovery rate (FDR; for W statistic)

5. Figure 2 shows that multiple knockoffs greatly improve power, but I am wondering whether it is because that the authors considered a small area (200 kb) with one causal region. The authors should compare different methods on a genome-wide scale with many casual regions.

Response: Thank you very much for the suggestion. In the previous single-region simulation study, multiple knockoffs greatly improve power at low target FDR. This is because using multiple knockoffs decreases the detection threshold from $1/q$ to $1/(Mq)$, where M is the number of

knockoffs. At higher target FDR (e.g. 0.20), multiple knockoffs and single knockoffs exhibit similar power for the same test (dispersion/burden).

In the newly added genome-wide & multiple disease loci simulation study using real data, we have further evaluated the power with different number of knockoffs at target FDR 0.10, where the detection threshold is no longer an issue. As expected, power is similar for different number of knockoffs. In conclusion, the power gain of multiple knockoffs is mainly due to different detection thresholds. Therefore, multiple knockoffs are particularly useful when the number of causal loci is small, and the target FDR is stringent.

Regardless of the effect on power, an important advantage in using multiple knockoffs is that it can significantly improve the stability and reproducibility of knockoff-based inference, as we discuss in the Methods section. Since the knockoff sample is random, each run of the knockoff procedure may lead to different selected sets of features. In practice, strong signals will always be selected but weak signals may be missed at random with a single knockoff. The proposed multiple knockoff procedure has significantly smaller variation in feature statistic in our simulation study based on real data from ADSP (Figure 9; Figure 3 in this letter).

Figure 3: Simulation studies to evaluate the stability and reproducibility of different knockoff procedures. Different colors indicate different knockoff procedures: *KnockoffScreen*, single knockoff and MK – Maximum (the multiple knockoff method based on the maximum statistic proposed by Gimenez and Zou²⁸). All three methods are based on the same knockoff generator proposed in this paper for a fair comparison. The stability is quantified as the variation of $\tau_{\Phi_{kl}}$ across 100 replicates due to randomly sampling knockoffs for a given data (left and right panels). The reproducibility is quantified as the frequency of a causal window being selected across 100 replicates.

6. *More investigation on the multiple knockoffs will be helpful. This paper considered five knockoffs. Do more knockoffs (more than 5) further increase the power? How can researchers decide the number of knockoffs?*

Response: As discussed above, we have evaluated the power with different number of knockoffs in the genome-wide & multiple disease loci simulation study. We observed that the power gain of multiple knockoffs is mainly due to different detection thresholds. Therefore, multiple knockoffs are particularly useful when the number of causal loci is small and the target FDR is stringent. Using more knockoffs won't provide additional increase in power once $M \geq 1/Nq$, where N is the number of independent causal variants and q is the target FDR. We have added a discussion on the practical choice of the number of knockoffs (Page 7, line 22).

7. *Population stratification adjustment: the authors compared the proposed method with*

unadjusted association tests. But the authors should compare it with PC-adjusted association tests, since no one uses association tests without PS adjustment (even when testing a seemingly homogeneous population). Also, the method cannot perfectly control PS as discussed by the author (cannot perfectly capture Z_i from G_i) and simulation results (Figure 4). Additionally, the method features causes difficulties in the PS adjustment. It uses a linear regression to generate G^{\sim} , so effectively assumes the same LD structure across individuals. The regression coefficients are not varying by populations. With different LD, exchangeability cannot be achieved. The author should clearly state that the method should be applied to the homogeneous population.

Response: Thank you very much for the suggestions. We have revised the section to better reflect what we have done, namely an *empirical* evaluation of *KnockoffScreen* in the presence of population stratification using real sequencing data from the ADSP project (Page 10, line 6). We also included comparisons with the conventional PC-adjusted association test as suggested. We observed that *KnockoffScreen* exhibits lower FDR than association test if they are unadjusted (which we agree is not what is done in practice), but both PC-adjusted *KnockoffScreen* and association test are able to control FDR, even though the generation of knockoffs does not explicitly model population structure. This is further illustrated by our real data analysis of ADSP where despite the combined analysis of three ethnicities there is no apparent inflation in false positive signals.

In addition, we have clarified that the current knockoff generator does not explicitly model heterogeneous LD structure across populations, and discussed the extension of knockoffs to studies with population structure (Page 11, line 9).

<https://www.biorxiv.org/content/10.1101/2020.08.04.236703v1.abstract>

8. *It isn't clear how PCs are adjusted in KnockoffScreen+10PCs.*

Response: We applied a conventional PC adjustment by including top 10 PCs as covariates when the p-values are calculated for each window. Note that the PC adjustment does not alter how the knockoffs are generated. We have clarified this in the main text accordingly (Page 10, line 18): “We also included a modified version of *KnockoffScreen* that adjusts for the top 10 global PCs when computing the p-values used to compute the window feature statistic, referred to as *KnockoffScreen+10PCs*.”

9. *In GWAS, it is common to have related individuals, but the method can(not) account for sample relatedness.*

Response: We agree that many large scale GWAS/WGS datasets have related individuals. Although the knockoff generator does not account for sample relatedness explicitly, since *KnockoffScreen* directly uses transformed p-values as importance score, it can serve as a wrapper method that can flexibly utilize p-values from methods that do model sample relatedness. We have added a discussion about potential applications of the proposed method to integrate newly proposed tests that account for sample relatedness (Page 18, line 31): “*KnockoffScreen* directly uses transformed p-values as importance score. This leads to another appealing property of *KnockoffScreen*, namely it can serve as a wrapper method that can flexibly utilize p-values from any existing or future association testing methods to achieve the benefits proposed here... To extend it to large-scale association studies with extremely unbalanced case-control ratios or sample

relatedness, one can apply methods like SAIGE⁴⁹ to calculate p-values for the original cohort and the synthetic cohort generated by *KnockoffScreen*, and then apply the same knockoff filter for variable selection.” It is of course of interest to extend the proposed knockoff generator to related individuals, but this is beyond the scope of the current paper.

10. Please report the computation time in ADSP and COPDGene data analysis.

Response: We have added a detailed discussion about the computation time (Page 19, Line 6): “We have demonstrated that the proposed sequential knockoff generator is significantly faster than existing alternatives. Besides the generation of knockoff features, another source of computational burden is the calculation of the importance score (p-value for a window). The total CPU time is 7,616 hours for the ADSP data analysis (15.2 hours with 500 cores) and 14,274 hours for the COPDGene data analysis (28.5 hours with 500 cores). The calculation of p-values in the current analysis is time consuming because of the comprehensive inclusion of many different functional annotations. Specifically, for each window, there are in total 29 tests being implemented for the original genetic variants and each of their five knockoffs, leading to a total of $29 \times 6 = 174$ p-value calculations per window. If computational resources are limited, using a limited number of functional annotations can substantially reduce the computing time. In addition, several methods have been proposed in recent years to use state-of-the-art optimization strategies for scalable association testing for large scale datasets with thousands of phenotypes in large biobanks. By directly utilizing p-values from those association testing methods, *KnockoffScreen* can scale up to biobank sized datasets at a comparable computational efficiency.”

Reviewer #3 (Remarks to the Author):

This paper describes a new method to identify putative causal loci (both rare and common variants) for a trait in whole-genome sequencing studies using recent advances in the theory of model-X knockoffs. The proposed method combines new ideas for generating genotype knockoffs that are appropriate for WGS studies with existing work on rare-variant tests and screening methods. The use of the knockoff framework brings with the advantages of a null that tests conditional independence i.e. is a locus associated with trait conditional on other loci. Thus, loci discovered using this approach are more likely to represent causal variants.

The paper is well-written with detailed and convincing experiments that demonstrate, for the most part, the utility of the proposed method over related approaches (notably, the approaches based on standard association tests and prior approaches for generating knockoffs). Overall, this work is a valuable contribution to the analysis of WGS studies. I do have a few major comments for the authors.

Response: Thank you very much for the nice summary of our paper, and the helpful suggestions. Below please find our point-by-point responses to your comments.

Major comments:

1. To show that *KnockoffScreen* is useful in prioritizing causal variants over variants that are in LD, the experiments should compare to methods explicitly designed for fine-mapping e.g. *CAVIAR* and/or *SUSIE*.

Response: Thank you for the suggestion. We want to clarify that the proposed method focuses on window-based analysis of whole genome sequencing data with rare variants, and we do not claim to be able to perform fine-mapping at the variant level. Fine mapping methods such as *CAVIAR* and *SUSIE* were developed for single variant analysis in array-based studies. To our knowledge, they are not applicable to the analysis of rare variants in sequencing studies. A recent paper introducing *KnockoffZoom* (focused on common variants in GWAS) has done comparisons with *CAVIAR* and *SUSIE*, but even *KnockoffZoom* cannot provide variant level resolution. Given these considerations and the fact that the primary goal of this paper is to provide a more powerful method for the analysis of whole-genome sequencing data, we have not included explicit comparisons, but we have added a discussion on this (page 2, line 8): “Moreover, a common feature of the existing association tests is that they often identify proxy variants that are correlated with the causal variants, rather than the causal variants that directly affect the traits of interest. Identification of putative causal variants usually requires a separate fine mapping step. However, fine mapping methods such as *CAVIAR* and *SUSIE* were developed for single, common variant analysis in GWAS studies, and are not directly applicable to the window-based analysis of rare variants in sequencing studies.”.

2. The robustness of *KnockoffScreen* to population stratification is interesting. Given that the method is particularly useful in the context of rare variants and that prior studies have hinted at differential confounding at rare variants that may not be adequately accounted for by principal components (Mathieson and McVean *Nature Genetics* 2012), testing the method under a model of differential confounding at rare vs common variants could further showcase the utility of the method.

Response: Thank you very much for the suggestion. To further evaluate whether *KnockoffScreen* can better control for population stratification involving rare variants, we have carried out additional simulation studies to simulate population stratification driven by rare variants using the ADSP data. Specifically, we randomly choose 100 regions across the whole genome but outside chromosome 19 with each region of size 200kb. Each region contains a 10kb causal window. We randomly set 10% rare variants (MAF<0.01; MAC>10) in each causal window to exhibit small effects on the trait of interest. Thus the allele frequency differences across ethnic groups will lead to different disease prevalence, reflecting population stratification driven by rare variants. Then we evaluate the FDR for the selected 200kb region near gene *APOE* (chr19: 44905796-44909393). Since the causal variants are independent of the target region, the confounding effect will be due to population stratification. Specifically, we generated the quantitative/dichotomous trait as follows:

$$\text{Quantitative trait: } Y_i = X_{i1} + \gamma \sum_{k=1}^{100} (\beta_{k1} g_{k1} + \dots + \beta_{k,k_s} g_{k,k_s}) + \varepsilon_i,$$

$$\text{Dichotomous trait: } g(\mu_i) = \beta_0 + X_{i1} + \gamma \sum_{k=1}^{100} (\beta_{k1} g_{k1} + \dots + \beta_{k,k_s} g_{k,k_s}) + \varepsilon_i,$$

where $X_{i1} \sim N(0,1)$, $\varepsilon_i \sim N(0,3)$ and they are all independent; (g_1, \dots, g_s) are selected risk variants; $g(x) = \log\left(\frac{x}{1-x}\right)$; for dichotomous trait, β_0 is chosen such that the prevalence is 10%. We set the effect $\beta_{kj} = \frac{a_k}{\sqrt{2m_{kj}(1-m_{kj})}}$, where m_{kj} is the MAF for the j -th variant in causal window k . We define a_k such that the variance due to the risk variants for each causal locus, $\beta_{k1}g_{k1} + \dots + \beta_{k,k_s}g_{k,k_s}$, is 0.01; we set $\gamma = 0, 0.25, 0.5, 0.75$ which quantifies the magnitude of population stratification.

We present the results in Table S1 (Table 1 in this paper). We found that both PC-adjusted *KnockoffScreen* and the conventional PC-adjusted association test are able to control FDR in the scenarios considered here, although *KnockoffScreen* exhibits a lower FDR for an unadjusted model.

Table 1: Empirical evaluation of *KnockoffScreen* in the presence of population stratification driven by rare variants. Each cell presents the empirical FDR. γ quantifies the magnitude of population stratification; C: continuous trait; D: dichotomous trait. *KnockoffScreen* controls FDR at 0.10. Association Testing is based on the usual Bonferroni correction (0.05/number of tests), controlling FWER at 0.05 level.

γ	Trait	KnockoffScreen	KnockoffScreen 10 PCs	Association Testing	Association Testing 10 PCs
0	C	0.098	0.102	0.022	0.020
0.25	C	0.084	0.096	0.094	0.024
0.5	C	0.124	0.084	0.430	0.018
0.75	C	0.196	0.068	0.926	0.028
0	D	0.106	0.112	0.056	0.058
0.25	D	0.108	0.100	0.184	0.042
0.5	D	0.198	0.110	0.846	0.030
0.75	D	0.312	0.090	0.996	0.034

3. *As the authors acknowledge, the procedure for generating knockoffs results in continuous-values genotypes. It is unclear how tests that require categorical genotypes are adapted to this setting (e.g. how would the burden test work in this setting?)*

Response: Most association tests such as dispersion (SKAT) or burden test can be directly applied to dosage data (i.e. continuous-value genotypes generated by imputation methods). We applied them in our setting in a similar fashion.

4. *I wonder what motivated the choice of the specific model for generating knockoffs. Since this model is not "exact" (as it generates continuous-valued genotypes), I presume the authors chose this for computational efficiency. Alternatives such as multinomial logistic regression model would produce categorical outcomes with additional computational costs. The authors should discuss this choice.*

Response: Thank you for the comment. In fact, we have previously evaluated a multinomial logistic regression model for generating categorical knockoffs. We found that the conditional mean of a rare variant can be extremely small, and it is very likely to generate knockoffs with all 0 values. The proposed algorithm is particularly designed to generate knockoff features for rare genetic

variants in sequencing studies, whose distribution is highly skewed and zero-inflated. It generates continuous variables to mimic genotype dosage value, making it more robust for rare variants. We have added a discussion about the multinomial logistic regression model (Page 20, line 30): “**This permutation-based algorithm is particularly designed to generate knockoff features for rare genetic variants in sequencing studies, whose distribution is highly skewed and zero-inflated. We note that the algorithm does not generate categorical variables in {0,1,2}. Instead, it generates continuous variables to mimic genotype dosage value, making it more robust for rare variants. In addition, we evaluated a multinomial logistic regression model for generating categorical knockoffs. We found that the conditional mean of a rare variant can be extremely small, and it is very likely to generate knockoffs with all 0 values where statistical inference cannot be applied.**”

5. *If the number of variants K is set to be $n^{1/3}$, then the runtime-per-variant would scale $O(n^{5/3})$ rather than $O(n)$.*

Response: Thank you for the careful review. We note that K is bounded by the size of the genomic region L , which does not increase as n increases. We have clarified this in the manuscript (Page 25, line 9): “ **L is a predefined constant for the length of the nearby region; K is the number of variants in the defined set B_j , which is bounded by the predefined constant L .**”

6. *Why is the target FDR different between African-Americans and non-Hispanic Whites?*

Response: We now report results at the same target FDR of 10% for all our analyses.

Reviewer #4 (Remarks to the Author):

This manuscript presents a novel method for testing variable importance in whole-genome sequencing studies using knockoffs, for the purpose of identifying likely causal variants while controlling the false discovery rate.

The statistical method of knockoffs \cite{candes2018} has been recently proposed as a solution to account for linkage disequilibrium in genetic studies \cite{sesia2019, sesia2020multi}. The key idea of that method is to augment the data with ‘dummy’ variables that behave in many ways similarly to the real variants, but are generated in silico by the statistician, and therefore are known to be non-causal and may be used as negative controls. Unlike permutation testing, knockoffs do not break the correlation structure among the variables, which is why they can account for linkage disequilibrium and thus help identify causal variants. This paper builds upon the framework of knockoffs, which previously focused on applications to SNP-array data from genome-wide association studies, and extends the existing methodology in different ways, focusing on the analysis of data from whole-genome studies (which include many more variants).

The main components in this paper are: (1) a new algorithm for generating knockoffs for whole-genome sequencing data; (2) the deployment within this context of an existing method for combining multiple knockoffs to improve the stability of the results; (3) applications to two whole-genome sequencing studies.

Overall, the manuscript is well-written and concerns a very relevant topic. In fact, it is true that new methods may be needed for the efficient analysis of whole-genome sequencing data, which are becoming increasingly available. These data raise new and interesting computational and statistical challenges compared to SNP arrays from genome-wide association studies, since they measure many more variants. In this sense, I think the authors are on the right track.

It is clear that a significant amount of work was put into this project, which I believe has the potential to lead to a good publication.

However, I have some doubts about important technical details of the proposed method that I would like to see addressed. I also have some objections about the accuracy of certain statements regarding earlier methods. I am aware that fully addressing my comments may require significant changes to the proposed method, but I am convinced that these changes could lead to a stronger publication.

Response: Thank you very much for the nice summary of our method.

`\section{Detailed feedback}`

`\subsection{Knockoff generator}`

This paper proposes a novel algorithm for generating knockoffs based on a modification of the Sequential Conditional Independent Pairs (SCIP) method in `\cite{candes2018}`. Unlike previous methods for generating knockoffs of genetic data, which were based first on a multivariate Gaussian approximation `\cite{candes2018}` and then on hidden Markov models `\cite{sesia2019,sesia2020multi}`, the approach proposed in this paper assumes variants more than 200kb apart to be independent of each other, and then models linearly the dependence of nearby variables. Two motivations are provided for this new method: faster computations, which can be useful for the analysis of whole-genome data, and the ability to easily obtain multiple knockoffs for each variant, which can improve the stability of the results `\cite{gimenez2019improving}`. However, I have some concerns about this proposal.

`\begin{enumerate}`

`\item \textbf{Interpretability.}` *As far as I can see, the proposed approach is not based on a well-defined model for the distribution of the genetic variants. I am afraid this makes it hard to understand under what assumptions is the validity of the inferences based. This approach is unlike that of much of the existing work on knockoffs `\cite{candes2018,sesia2019,gimenez2019improving,bates2020metropolized}`, which is based on well-defined models. In*

particular, since no model is explicitly assumed for the joint distribution of G , it is not clear under what assumptions could the exchangeability (in the knockoff sense) of G and \tilde{G} be rigorously proved. I do not find the argument presented in the manuscript to justify the correctness of the knockoff construction to be fully satisfactory, as it appears to me to be somewhat circular.

Response: Thank you for these insightful remarks. Indeed, the main motivation for developing the new knockoff generator are computational efficiency and ability to generate knockoff variable for the rare variants that enjoy the necessary exchangeability properties. While the existing HMM-based model is commonly used to model genetic data, it is very computationally expensive for unphased data (which is the usual format for sequencing studies), and furthermore it does not guarantee the exchangeability properties for rare variants. We now provide a theoretical argument for a simplified, but a good approximation of the real data.

The knockoff generator proposed in this paper is based on the observation that the correlation among genetic variants approximately exhibits a block diagonal structure (e.g. Gabriel et al. Science 2002) due to the nature of linkage disequilibrium in a genetic region. The genotypes are modeled similarly as dosage values (i.e. continuous-value genotypes generated by imputation methods). In particular, if a multivariate normal distribution with block diagonal covariance matrix is assumed to model the genotypes, we can derive the following useful results: when we apply Algorithm 1 to generate the knockoff variables, for the j th variant, the conditional distribution $\mathcal{L}(G_j | G_{-j}, \tilde{G}_{1:(j-1)})$ is fully determined by those variants belonging to the same block and their already generated knockoffs up to the current step; the conditional distribution is still normal with the mean being the linear combination of those variables.

We design the knockoff generator based on these important observations. When sampling from the conditional distribution, we permute the residuals instead of directly generating samples from a normal distribution. Based on our simulation studies, by permuting the residuals the exchangeability can be better preserved, especially for the rare variants, whose distribution is highly skewed and zero-inflated. We also provide more details of the theoretical justification of the knockoff generator in the Appendix.

Non-linear dependencies. *Linear dependencies may not be sufficient to realistically describe the distribution of genetic variants, especially when rare variants are involved. This is why the standard models adopted in the literature for population genetics or phasing/imputation are based on hidden Markov models. For example, suppose that each individual in a population must have one of the following 4 haplotypes:*

$$\begin{aligned} & G_1 & G_2 & G_3 & | & 1 & 0 & 1 & | & 1 & 1 & 0 & | & 0 & 1 & 1 & | & 0 & 0 & 0 \end{aligned}$$

Clearly, G_3 is not independent of G_1, G_2 (knowing both G_1 and G_2 can be used to predict G_3 exactly); however, it is linearly uncorrelated with both of them. The method proposed in this paper would treat G_3 as independent of G_1 and G_2 , which does not account correctly for linkage disequilibrium in this example.

It is well known that the magnitude of linear correlations is affected by the allele frequency of the variants \cite{wray2005allele}, so I am afraid that the proposed approach may not accurately capture linkage disequilibrium especially for rarer variants.

Response: We agree with the reviewer that a linear model may not fully characterize the conditional distribution and then the joint distribution. The current choice of a linear model is mainly due to the computational efficiency. However, the model can be further modified to incorporate non-linear effects using more sophisticated learning methods, as discussed on Page 21, line 22.

We have added an empirical evaluation of the methods and HMM-based knockoff generators (with $S=12$ and $S=50$ states), stratified by allele frequency. We present the results in Figures S6 and S7. As shown, the proposed method generates knockoff versions for rare variants with better exchangeability with the original variants compared with the HMM model. That is, the correlation coefficients are closer to those of the original variants for *KnockoffScreen* compared to HMM (bottom panel, the dots are mostly above the diagonal line). In addition, we have updated Figure 2 in the main text to include the HMM with $S=50$ states and the pattern remains similar.

One plausible explanation is that the application of HMM to whole genome sequencing data requires accurate phased data for rare variants, which itself is a challenging task and also an active research area.

Figure 4: Comparison with HMM (S=12) stratified by minor allele frequency. We generated 10,000 individuals with genetic data for a 200 kb region containing 1000 genetic variants, simulated using a coalescent model (COSI). We compared the proposed algorithm to HMM with number of states S=12 and evaluated whether the second order (correlation between each pair of genetic variants) is exchangeable. Each dot presents one variant/window. The left panels evaluate how the correlation structure of knockoffs is similar to that of the original variants; the right panels evaluate how the knockoffs preserve the correlation structure when one swaps a variant with its synthetic counterpart.

Figure S7: Comparison with HMM (S=50) stratified by minor allele frequency. We generated 10,000 individuals with genetic data for a 200 kb region containing 1000 genetic variants, simulated using a coalescent model (COSI). We compared the proposed algorithm to HMM with number of states S=50 and evaluated whether the second order (correlation between each pair of genetic variants) is exchangeable. Each dot presents one variant/window. The left panels evaluate how the correlation structure of knockoffs is similar to that of the original variants; the right panels evaluate how the knockoffs preserve the correlation structure when one swaps a variant with its synthetic counterpart.

\item \textbf{Long-range dependencies.} *Variants 200kb apart are not necessarily independent. Even in homogeneous populations, where population structure is relatively weak, linkage disequilibrium can cause spurious discoveries much beyond 200 kb from the nearest causal variant \cite{sesia2020multi}. If the population is stratified or admixed, dependencies have much longer range, to the point that not even variants on different chromosomes can be said to be independent \cite{sesia2020controlling, bates2020causal}.*

Response: The conditional auto-regressive model assumes an auto-regressive covariance structure,

similar to a Markov model. It indeed assumes a decayed correlation as the distance increases and thus only accounts for local LD structure. The typical LD block is less than 100 kb. LD within a block decreases slowly, but between blocks the decay is rapid (e.g. Anderson and Novembre, AJHG, 2003). Therefore, we agree with the reviewer that the current method cannot capture strong long-range LD due to population stratification. We have added a discussion about this limitation and cite the paper *sesia2020controlling* for future extension of the method (Page 11, line 9): “However, we clarify that *KnockoffScreen* itself does not completely eliminate the confounding due to population stratification (Table S1) because the current knockoff generator assumes the same LD structure across individuals and it only accounts for local LD structure. Therefore, it does not capture heterogeneous LD structure across populations and strong long-range LD due to population stratification. Development of new knockoff generators that explicitly account for population structure will be of interest.”

\item \textbf{Rare variants.} It is not entirely clear why the proposed approach is better suited to describe the distribution of rare variants compared to hidden Markov models, which are the standard probabilistic model used by many state-of-the-art phasing methods for whole-genome sequence data \cite{delaneau2019accurate}. Hidden Markov models can account quite naturally for rare variants, whereas linear correlations may not correctly account for LD between variants with very different allele frequencies \cite{wray2005allele}.

Response: We would like to clarify that the proposed method is an alternative knockoff generator that uses unphased genotype data, while the HMM first estimates the haplotypes and then generates the knockoffs. Since it is difficult to accurately phase haplotypes for rare variants, this may impact the performance of the HMM-based knockoff generation. As mentioned previously, we have empirically evaluated the proposed method and the HMM method in terms of the exchangeability of the resulting knockoffs with the original variables for rare variants, and have shown better performance for the proposed method.

\item \textbf{Computational cost.} In theory, the proposed knockoff generator has the same computational complexity as the existing methods based on hidden Markov models. The authors mention that their implementation of the proposed method is faster than the current implementation of the state-of-the-art method for hidden Markov model knockoffs, citing \cite{sesia2019}. However, the state-of-the-art implementation is that of \cite{sesia2020multi}, which is orders of magnitude faster than that of \cite{sesia2019}, and it has already been applied to much larger data sets than those considered here.

Response: Thank you for your comment. We have in fact followed *\cite{sesia2020multi}* for implementing the HMM method, as described in Supplemental Figure S1 of *\cite{sesia2020multi}*, i.e. use fastPhase to fit an HMM model and then SNPknock to generate the knockoffs. We cited *\cite{sesia2019}* only because it is the original paper proposing the HMM model. We have now revised the citation.

We note that the proposed method focuses on the analysis of whole genome sequencing data, and thus the computational cost is measured on unphased data, which is the usual format for sequencing

studies. We have clarified that the computational cost of HMM can be potentially improved by a pre-processing step using more advanced phasing methods for whole-genome sequence data.

Furthermore, we have expanded the evaluation of the computational cost to include HMM with $S=12$ and $S=50$ states, and record the computing time for fastPhase and SNPknock separately. The results are summarized in Table 1 (Table 2 in this letter). We agree with the reviewer that the computational complexity of HMM is of the same order as the proposed method (linear in n and p), as shown in the table below. Both methods work better than the second-order knockoff generator when p is sufficiently large. However, the time spent for each fixed n and p is significantly longer for the HMM method than for the proposed method.

In addition, we want to clarify that the whole genome sequencing dataset considered in this paper ($n \times p \sim 4,000 \times 80,000,000 = 3.2e+11$) is not smaller than the biobank datasets considered in \cite{sesia2020multi} ($n \times p \sim 400,000 \times 600,000 = 2.4e+11$).

Table 2: Computing time of different knockoff generators. Each cell shows the computing time in seconds to generate knockoffs based for unphased genotype data. The multiple sequential knockoffs approach generates five knockoffs. The computing time was measured on unphased genotype data using a single CPU (Intel(R) Xeon(R) CPU E5-2640 v3 @ 2.60GHz). Since the HMM model was mainly proposed for phased data, we report the computing time separately for phasing with fastPhase and sampling with SNPknock.

n	p	MSK (5 knockoffs)	SK	SecondOrder	HMM with S=12		HMM with S=50	
					Phasing	Sampling	Phasing	Sampling
1000	500	2.11	0.86	8.9	37.86	6.02	580.87	93.88
1000	1000	3.99	1.92	57.01	76	12.01	1147.66	188.74
1000	2000	8.89	4.06	491.19	161.94	24.76	2336.83	376.93
5000	500	4.66	1.63	8.51	188.5	30.45	2878.43	485.34
5000	1000	11.76	3.95	52.63	380.06	60.28	5914.19	996.11
5000	2000	31.58	11.09	479.01	811.61	129.6	11734.66	1865.11
10000	500	7.42	2.34	9.29	377.07	58.8	5784.24	957.49
10000	1000	20.57	6.59	54.66	757.49	123.94	11744.68	1936.85
10000	2000	52.86	16.92	445.05	1571.19	253.46	23584.8	3870.07

\end{enumerate}

\subsection{Group-wise testing}

*The authors mention the need to modify the original knockoff generation algorithm to avoid having zero power in the presence of tightly linked variants. They cite the variant pruning approach in \cite{candes2018, sesia2019} as the current state-of-the-art to deal with this issue, and then propose an alternative solution based on group-wise exchangeable knockoffs. However, the current state-of-the-art in dealing with knockoffs for tightly linked variants is that of \cite{sesia2020multi}, which proposes a solution similar to that advanced here. Despite similar ideas, the method proposed in this paper is not fully correct. The problem in the approach proposed here is that the knockoffs for G_1 and G_2 should be generated *jointly* given*

G_3 to ensure that the correct exchangeability holds (example on page 23); see \cite{sesia2020multi}.

On page 23, the authors acknowledge that their solution is not rigorous, but they incorrectly state the problem is still unsolved in theory. In truth, the problem has been solved in the work of \cite{sesia2020multi}, of which the authors should be aware because they cite it elsewhere in their manuscript.

Response: Thank you for the helpful comments. We have revised the discussion to reflect the state-of-the-art way to deal with this issue. We also appreciate your suggestions to revise the algorithm. We have revised the method accordingly. Our modified algorithm now simultaneously generates \tilde{G}_1 and \tilde{G}_2 based on a joint distribution $P(G_1, G_2|G_3)$, by first estimating the conditional means and then permuting the residuals jointly. We have generated all the results in the revised manuscript using this modified version.

\subsection{Population structure}

It is true that knockoffs reduce false positives due to unadjusted population stratification, as mentioned by the authors and previously discussed in \cite{sesia2019, sesia2020multi}. However, this is only true if the knockoffs are correctly exchangeable with the real variants, which depends on how they are constructed. The construction proposed here treats variants that are more than 200kb apart as independent; therefore, it does not seem to account for those long-range dependencies that characterize population structure. See \cite{sesia2020multi, sesia2020controlling} for a more complete discussion of this issue.

In light of this, I am not sure it is correct to say the proposed method can account for population structure in principle. Furthermore, this manuscript suggests using marginal association tests, which are not as robust to population structure as multivariate tests \cite{klasen2016}. That being said, it is true that regressing out the top principal components while computing the association statistics probably mitigates these weaknesses.

Response: We have revised the section to better reflect what we have done, namely an *empirical* evaluation of *KnockoffScreen* in the presence of population stratification using real genotype data from the ADSP project. We also included conventional PC-adjusted association test for comparison as suggested. We observed that *KnockoffScreen* exhibits lower FDR than association test if they are unadjusted (which we agree is not what is done in practice), but both PC-adjusted *KnockoffScreen* and association test are able to control FDR, even though the generation of knockoffs does not explicitly model population structure. This is further illustrated by our real data analysis of ADSP where despite the combined analysis of three ethnicities there is no apparent inflation in false positive signals. We also cite the paper *sesia2020controlling* and clarify that *KnockoffScreen* itself does not completely eliminate the confounding due to population stratification (Table S1) because the current knockoff generator assumes the same LD structure across individuals and it only accounts for local LD structure. Therefore, it does not capture heterogeneous LD structure across populations and strong long-range LD due to population

stratification. Development of new knockoff generators that explicitly account for population structure will be of interest (Page 11, Line 9).

\subsection{Association tests}

Even though the framework of knockoffs is sufficiently general to accommodate different association tests \cite{candes2018}, this manuscript seems to advocate for the use of marginal association tests in order to localize causal variants.

I find this approach a little counter-intuitive. If the goal is to distinguish between causal variants and variants that are only marginally associated with the phenotype, marginal statistics are not the most powerful option. In fact, the most effective approaches for fine-mapping are multivariate; see for example \cite{wang2019simple} and references therein. Even genome-wide testing for SNP array data is now carried out with multivariate statistics by default (linear mixed models), because these are much more powerful and robust to population structure compared to marginal testing. From a statistical perspective, it is well-known that marginal statistics are less powerful than multivariate statistics unless the phenotype is monogenic, which is not the case for the applications considered in this paper.

This is why knockoffs were previously applied in combination with multivariate statistics \cite{candes2018, sesia2019, sesia2020multi}. It is well known that knockoffs can also be applied with univariate statistics, but such approach should be expected to be less powerful and robust \cite{sesia2020controlling} compared to multivariate testing.

I recognize some interesting advantages of univariate statistics: namely, their lower computational cost and their ease of use in combination with multiple knockoffs. However, more work may be needed to justify the trade-off, especially since efficient implementations of more powerful multivariate statistics may be possible even for full genome sequencing data.

Response: We agree with the reviewer that the multivariate model has many advantages over marginal association testing, which is suboptimal but widely applied in the GWAS analyses due to its efficient implementation. We note that our approach is not limited to univariate statistics, and p-values from multivariate models (e.g. BOLT-LMM, and its extension to region-based analysis of sequencing data) can be provided as input to our current method. We have added a discussion about potential applications of the proposed method to integrate tests based on multivariate model (page 18, line 38): “Moreover, recent studies have demonstrated that multivariate models have many advantages over marginal association testing, including improved power by reducing the residual variation and better control of population stratification¹⁵. *KnockoffScreen* is able to integrate tests from multivariate models (e.g. BOLT-LMM and its extension to window-based analysis of sequencing data)”

We consider the proposed method to be complementary to lasso-based knockoff inference (e.g. KnockoffZoom), and we think this is only the first step in bridging state-of-the-art region-based association tests for sequencing data with knockoff statistics for putative causal window discovery.

We hope that the flexibility of the proposed framework will also facilitate further methodological developments and wider application of knockoff statistics in genetic studies.

\subsection{Numerical experiments}

The simulations on page 22 are intended to mimic the real data analysis of ADSP. However, ADSP is a polygenic disease, whereas there is only one causal locus in this simulation (we set 1.25% variants to be causal, all within a 5kb signal window). I am not fully convinced by this design, for the reasons outlined below.

\begin{enumerate}

I fear that having only one causal locus may partially hide the loss in power resulting from the choice of using univariate rather than multivariate test statistics. This is an important trade-off that I feel should be discussed more carefully.

Response: We have expanded our simulations by using genome-wide data from the ADSP project and assuming multiple causal regions. Specifically, we randomly choose 10 causal loci and 500 noise loci across the whole genome with each spanning 200kb. Each causal locus contains a 10kb causal window. For each replicate, we randomly select 10% variants in each 10kb window to be causal. Across the genome there are on average 335 causal variants per replicate. We compared to state-of-the-art region-based methods for sequencing studies adjusted by Bonferroni correction for FWER control, and BH procedure for FDR control, and evaluated different number of knockoffs. We have added a new section to present these results (page 7, line 5).

The simulation results show that *KnockoffScreen* exhibits substantially higher power than Bonferroni correction, consistent with the real data analysis where *KnockoffScreen* is able to identify many disease-associated loci missed by Bonferroni correction. Conventional BH FDR control fails to control FDR at the target level due to the complicated correlation structure among genetic variants.

We have also acknowledged the advantages of multivariate model over marginal association testing. We have added a discussion about potential applications of the proposed method to integrate new region-based tests based on multivariate model (page 18, line 38).

I am not sure the comparison with the existing knockoff methods is as fair as it could be, given that those are designed to work genome-wide for the analysis of polygenic traits. These methods are already known to be relatively powerless/unstable when all causal variants are in a single locus (they require at least 10 distinct discoveries when applied at FDR level 10%, as correctly pointed out by the authors in this manuscript), but I would expect them to be generally more powerful for polygenic traits

Response: Thank you very much for the comment. In the previous single-region simulation study, multiple knockoffs greatly improve power at low target FDR. This is because using multiple knockoffs decreases the detection threshold from $1/q$ to $1/(Mq)$, where M is the number of knockoffs. At higher target FDR (e.g. 0.20), multiple knockoffs and single knockoffs exhibit similar power for the same test (dispersion/burden).

In the newly added genome-wide & multiple disease loci simulation study using real data, we have further evaluated the power with different number of knockoffs at target FDR 0.10, where the detection threshold is no longer an issue. As expected, power is similar for different number of knockoffs. In conclusion, the power gain of multiple knockoffs is mainly due to different detection thresholds. Therefore, multiple knockoffs are particularly useful when the number of causal loci is small, and the target FDR is stringent. More knockoffs won't further increase the power once $M \geq 1/Nq$, where N is the number of independent causal variants and q is the target FDR. We have added a discussion on the practical choice of the number of knockoffs.

Regardless of the effect on power, an important advantage in using multiple knockoffs is that it can significantly improve the stability and reproducibility of knockoff-based inference, as we discuss in the Methods section. Since the knockoff sample is random, each run of the knockoff procedure may lead to different selected sets of features. In practice, strong signals will always be selected but weak signals may be missed at random with a single knockoff. The proposed multiple knockoff procedure has significantly smaller variation in feature statistic in our simulation study based on real data from ADSP (Figure 9; Figure 6 in this letter).

Figure 6: Simulation studies to evaluate the stability and reproducibility of different knockoff procedures. Different colors indicate different knockoff procedures: *KnockoffScreen*, single knockoff and MK – Maximum (the multiple knockoff method based on the maximum statistic proposed by Gimenez and Zou²⁸). All three methods are based on the same knockoff generator proposed in this paper for a fair comparison. The stability is quantified as the variation of $\tau_{\Phi_{kl}}$ across 100 replicates due to randomly sampling knockoffs for a given data (left and right panels). The reproducibility is quantified as the frequency of a causal window being selected across 100 replicates.

item This simulation may not a very realistic representation of an efficient analysis insofar as it does not leverage the information about the phenotype contained in farther apart loci. By contrast, it would be more efficient to use multivariate statistics, such as linear mixed models (e.g., BOLT-LMM), Bayesian multivariate regression models (e.g., SUSIE), or sparse regression models (e.g., the lasso), since these can explain a larger fraction of the variance in the phenotype.
 \end{enumerate}

Response: The proposed method is a region-based scanning method for rare variants analysis in sequencing studies. Therefore, we compared it with state-of-the-art methods for sequencing studies (e.g. SKAT, burden, and ACAT), instead of methods designed for SNP-array data including BOLT-LMM and fine-mapping methods such as CAVIAR and SUSIE. Extending region-based tests for sequencing data analysis to multivariate model is a non-trivial and active research area. We have added a discussion about the extension of the proposed method to multivariate model to achieve the aforementioned advantages (Page 18; line 38).

Lee, S., Abecasis, G.R., Boehnke, M. & Lin, X. Rare-variant association analysis: study designs and statistical tests. *The American Journal of Human Genetics* **95**, 5-23 (2014).

Liu, Y. *et al.* Acat: A fast and powerful p value combination method for rare-variant analysis in sequencing studies. *The American Journal of Human Genetics* **104**, 410-421 (2019).

REVIEWERS' COMMENTS

Reviewer #1 (Remarks to the Author):

The authors relatively well addressed my comment. I have one minor comment:

Figure 3 shows that conventional FDR control (BH, Grey) had much higher power than the proposed approach. With 200 kb resolution, BH-FDR is slightly inflated in Quantitative traits and well-controlled in binary traits. In the manuscript, "Conventional BH FDR control fails to control FDR at the target level due to the complex correlation structure among genetic variants/windows", but this isn't correct, as FDR is relatively well controlled with 200 kb resolution. The author should acknowledge that the power can be reduced with the Knockoffscreen. I think the strength of Knockoffscreen is its ability to distinguish the causal signals (as well summarized in Abstract) but given the same FDR control, I don't think it can improve power over (conventional) marginal tests.

Reviewer #3 (Remarks to the Author):

The authors have addressed my concerns. I am happy to support publication.

Reviewer #4 (Remarks to the Author):

I thank the authors for providing many clarifications and additional simulations to address my previous comments.

As far as one could achieve without substantially altering the proposed method, I believe they have addressed all my concerns.

Nonetheless, I am still a little concerned about what I see as fundamental statistical limitations of the proposed approach:

- Univariate methods may be computationally convenient but they are not designed to discover causal variants, a stated goal of this paper. Unfortunately, LMMs are only partly multivariate, in the sense that they do not account for the effects of multiple variants on the same chromosome. Therefore, even if applied with LMM-based instead of univariate p-values, the proposed method still seems a bit unsatisfactory compared to fully multivariate fine-mapping or knockoff methods.
- The linear approximate model used to generate knockoffs is computationally convenient, but it is neither very interpretable nor very realistic.

We thank the reviewers for their insightful comments. In the revised version, we have made further modifications to the main text and expanded upon the limitations in the Discussion section to address the remaining comments.

Reviewer #1 (Remarks to the Author):

The authors relatively well addressed my comment. I have one minor comment:

Figure 3 shows that conventional FDR control (BH, Grey) had much higher power than the proposed approach. With 200 kb resolution, BH-FDR is slightly inflated in Quantitative traits and well-controlled in binary traits. In the manuscript, “Conventional BH FDR control fails to control FDR at the target level due to the complex correlation structure among genetic variants/windows”, but this isn’t correct, as FDR is relatively well controlled with 200 kb resolution. The author should acknowledge that the power can be reduced with the KnockoffScreen. I think the strength of KnockoffScreen is its ability to distinguish the causal signals (as well summarized in Abstract) but given the same FDR control, I don’t think it can improve power over (conventional) marginal tests.

Response: Thank you very much for the helpful comments. We have rephrased the conclusion to better summarize the simulation results (page 7, line 18): “The simulation results show that *KnockoffScreen* exhibits substantially higher power than using Bonferroni correction. Additionally, using the conventional BH FDR may have higher power than *KnockoffScreen*, but fails to control FDR at higher resolution (e.g. +/-75 kb).”

In the case of independence among tests, the FDR control with BH procedure and that with knockoffs are expected to have similar power and FDR. With correlated genetic variants/windows, the higher power of BH FDR control observed in the simulation studies can lead to false positive inflation due to complex correlation structure. Therefore, we do not recommend directly using the BH FDR control in whole genome sequencing studies. We have added a discussion about the power (page 7, line 20).

In addition, we have now added the BH FDR control to our ADSP analysis and report the results in Supplemental Figure S6. As an example, for ADSP (Figure 1 in this letter) we observed that both BH FDR control and *KnockoffScreen* identified the strong signals at the APOE locus. The additional associations (e.g. *KAT8*) identified by *KnockoffScreen* are missed by the BH FDR control, but they are likely true associations as they were previously reported in the GWAS catalog to be associated with AD. On the other hand, most additional associations identified by BH FDR control were not reported by previous GWAS studies, and may be false positives.

Figure 1: The analysis of the Alzheimer’s Disease Sequencing Project (ADSP) data with the Benjamini–Hochberg procedure for FDR control. The left panel presents the Manhattan plot of adjusted p-values (Q-values; truncated at 10^{-10} for clear visualization) from the conventional association testing with the Benjamini–Hochberg adjustment for FDR control. The right panel presents a heatmap that shows stratified p-values (truncated at 10^{-10} for clear visualization) of all loci passing the FDR=0.1 threshold, and the corresponding adjusted p-values that already incorporate correction for multiple testing. For each locus, the adjusted p-values of the top associated single variant and/or window are shown indicating whether the signal comes from a single variant, a combined effect of common variants or a combined effect of rare variants. The names of those genes previously implicated by GWAS studies are shown in bold (names were just used to label the region and may not represent causative gene in the region).

Reviewer #4 (Remarks to the Author):

I thank the authors for providing many clarifications and additional simulations to address my previous comments. As far as one could achieve without substantially altering the proposed method, I believe they have addressed all my concerns.

Nonetheless, I am still a little concerned about what I see as fundamental statistical limitations of the proposed approach:

- *Univariate methods may be computationally convenient but they are not designed to discover causal variants, a stated goal of this paper. Unfortunately, LMMs are only partly multivariate, in the sense that they do not account for the effects of multiple variants on the same chromosome. Therefore, even if applied with LMM-based instead of univariate p-values, the proposed method still seems a bit unsatisfactory compared to fully multivariate fine-mapping or knockoff methods.*
- *The linear approximate model used to generate knockoffs is computationally convenient, but it is neither very interpretable nor very realistic.*

Response: Thank you very much for the helpful comments. We have expanded upon the limitations in the Discussion section (page 13, line 19) as follows: “Despite the aforementioned advantages, *KnockoffScreen* has some limitations related to underlying modeling assumptions needed to improve the computational efficiency of the multiple knockoff generation and calculation of the

feature importance scores. In particular, the implemented feature importance scores rely on computing p-values from a marginal model (e.g. single variant score test, burden test or SKAT) or a partly multivariate model (BOLT-LMM and its extension to window-based analysis of sequencing data). We made this choice of feature importance score due to its flexibility to integrate state-of-the-art tests for sequencing studies, but we recognize that a fully multivariate model as implemented in Sesia et al. (2020) can be more powerful. In addition, the knockoff generator used in *KnockoffScreen* assumes a linear approximate model based on unphased genotype dosage data. This model is well motivated based on the sequential model to generate knockoff features, and the approximate multivariate normal model for the genotype data commonly used in the genetic literature. Additionally, it is computationally efficient relative to existing knockoff generation methods. We acknowledge that relative to a generative model like HMM it is less interpretable. More complex models for discrete genotype values that can also account for non-linear effects among genetic variants could be of interest in future work.”

Sesia, M., Katsevich, E., Bates, S., Candès, E. & Sabatti, C. Multi-resolution localization of causal variants across the genome. *Nature Communications* 11, 1093 (2020).